# DNALongBench: A Benchmark Suite for Long-Range DNA Prediction Tasks

## Abstract

Modeling long-range DNA dependencies is crucial for understanding genome structure and function across a wide range of biological contexts in health and disease. However, effectively capturing the extensive long-range dependencies between DNA sequences, spanning millions of base pairs as seen in tasks such as three-dimensional (3D) chromatin folding, remains a significant challenge. Additionally, a comprehensive benchmark suite for evaluating tasks reliant on long-range dependencies is notably absent. To address this gap, we introduce DNALongBench, a benchmark dataset spanning five important genomics tasks that consider long-range dependencies up to 1 million base pairs: enhancer-target gene interaction, expression quantitative trait loci, 3D genome organization, regulatory sequence activity, and transcription initiation signal. To comprehensively assess DNALongBench, we evaluate the performance of five baseline methods: a task-specific expert model, a convolutional neural network (CNN)-based model, and three fine-tuned DNA foundation models – HyenaDNA, Caduceus-Ph and Caduceus-PS. We envision DNALongBench having the potential to become a standardized resource that facilitates comprehensive comparisons and rigorous evaluations of emerging DNA sequence-based deep learning models that consider long-range dependencies.

## 1 Introduction

Genomic DNA sequences are the blueprint of life, guiding the development of cellular complexities. Although DNA sequences encoding proteins are responsible for the diverse biochemical functions within organisms, it is noteworthy that most eukaryotic genomes consist of non-coding sequences interspersed with protein-coding regions. These non-coding sequences contain a variety of gene regulatory elements, including promoters, enhancers, non-coding RNAs, and other functional non-coding elements that orchestrate when and where genes are turned on and off. Over the past two decades, large-scale functional genomic projects, such as the ENCODE project (Consortium et al., 2012), have successfully cataloged a vast collection of putative non-coding regulatory elements in the human genome. However, despite these advances, our understanding of how these elements regulate gene expression remains limited. One critical challenge is that genomes are dynamically folded into multi-scale 3D structures inside the cell nucleus, resulting in widespread physical DNA-DNA interactions, even between regions located megabases apart (Dekker & Misteli, 2015; Furlong & Levine, 2018; Zhang et al., 2024). Effectively determining which of these interactions are functionally relevant to cellular processes across different biological contexts requires significant experimental effort.

To address this challenge, the increasing availability of genomic data, such as ChIP-seq (Furey, 2012), ATAC-seq (Klemm et al., 2019), Hi-C and its derivatives (Kempfer & Pombo, 2020), has led to the development of supervised deep learning methods that show great promise in systematically delineating sequence-to-function relationships. For instance, convolutional neural networks (CNNs) and transformer-based methods have proven effective in characterizing regulatory elements (Zhou & Troyanskaya, 2015; Alipanahi et al., 2015; Quang & Xie, 2016; Avsec et al., 2021b), predicting spatial proximity between genomic loci (Fudenberg et al., 2020; Schwessinger et al., 2020), and predicting gene expressions from local sequence context (Avsec et al., 2021a). Despite these advancements, capturing the dependencies across very long distal DNA elements remains computationally challenging due to the scarcity of experimental data and the difficulty in modeling long-range sequence dependencies (Karollus et al., 2023).

Recently, large language models (LLMs) have revolutionized the natural language processing (NLP) field, demonstrating remarkable capabilities across a broad spectrum of applications (Devlin et al., 2018; Wei et al., 2022; Achiam et al., 2023; Touvron et al., 2023). These models first leverage self-supervised learning techniques to learn the intricate patterns from vast amount of unlabeled text data, followed by fine-tuning steps tailored to specific tasks. Recognizing structural similarities between DNA sequences and natural language (Tang & Koo, 2024), several DNA foundation models have emerged (Nguyen et al., 2024a;b; Schiff et al., 2024). However, the advantages of these models in addressing meaningful biological questions are still controversial, leaving a critical question unsolved: *Could foundation models pre-trained on genomic DNA sequences offer a new paradigm shift in understanding the interactions between regulatory elements and genes?* Answering this question requires evaluating models on benchmark datasets to assess their performance, identify limitations, and drive future improvements. However, most foundation models pre-trained on genomic DNA sequences have so far only been evaluated on prediction tasks involving sequences up to thousands of base pairs in lengths, such as the identification of regulatory elements and gene expression prediction (Marin et al., 2023; Grešová et al., 2023; Dalla-Torre et al., 2023; Zhou et al., 2023; Kao et al., 2024). The potential of DNA LLMs to capture long-range interactions between DNA sequences in various contexts have not been well evaluated.

Here, we introduce DNALONGBENCH, the largest collection to date of realistic and biologically meaningful genomic DNA prediction tasks that require long-range sequence input and involve long-range dependencies. DNALONGBENCH comprises five different tasks and datasets, each covering critical aspects for studying gene regulation at various length scales. The contributions of DNALONGBENCH are three-fold:

- We introduce DNALONGBENCH, a benchmark for long-range DNA prediction tasks spanning up to 1 million base pairs (bp) across five distinct tasks. To the best of our knowledge, DNALONGBENCH is the most comprehensive benchmark tailored towards long-range DNA prediction tasks available to date.
- We evaluate the proposed DNALONGBENCH using three representative models, demonstrating that while DNA foundation models can capture long-range depdencies to certain extent, the expert models consistently outperform DNA foundation models across all five tasks.
- The models exhibit varying performance across different tasks, highlighting the diverse challenges inherent in the DNALONGBENCH prediction tasks and revealing the differing levels of difficulty associated with each task.

We envision DNALONGBENCH as a valuable resource for evaluating foundation models trained on DNA sequences with a focus on the capabilities of modeling long-range genomic interactions. Code and Data of DNALONGBENCH are available at https://anonymous.4open.science/r/DNALongBench-FB1D. We also provide a leaderboard at https://dnalongbench.github.io/DNALongBench.

## 2 RELATED WORK

### 2.1 EXISTING BENCHMARK DATASETS CONSIDERING LONG-RANGE DNA SEQUENCE

The benchmark datasets specifically designed to evaluate the capabilities of DNA foundation models in capturing long-range DNA dependencies remain underexplored. Most existing benchmarks for DNA foundation models primarily focus on short-range tasks (e.g., thousands of base pairs) and binary classification. To date, BEND (Marin et al., 2023) and the Genomics Long-range Benchmark (LRB) (Kao et al., 2024) are the only two existing benchmark datasets that include long-range genomic DNA prediction tasks. BEND comprises two long-range tasks: enhancer annotation and gene finding, both involving the classification of regulatory elements. LRB, on the other hand, adapted all their tasks from the Enformer (Avsec et al., 2021a) paper and curated three datasets focused on gene expression prediction and the effects of variants on gene expression. Notably, both BEND and LRB are limited in scope, focusing specifically on the identification of regulatory elements or gene expression-related prediction, and thus overlook other important long-range DNA prediction tasks. For example, neither benchmark includes structure-related predictive tasks requiring ultra-long sequences, such as contact map prediction and enhancer-target gene prediction. Furthermore, they lack base-pair-resolution regression tasks for quantitative assays. As a result, a comprehensive benchmark suite for evaluating a broader range of tasks reliant on long-range dependencies is still absent. We compare the scope of previous benchmarks for DNA prediction tasks with DNALONGBENCH in **Table** 1.

| Benchmark Feature | Genomic Benchmarks | NT Benchmark | GUE | BEND | LRB | DNALONGBENCH |
|---|---|---|---|---|---|---|
| Has Long-range Task | × | × | × | ✓ | ✓ | ✓ |
| Longest Input (bp) | 4,707 | 600 | 1,000 | 100k | 192k | 1M |
| Has Base-pair-resolution Regression Task | × | × | × | × | × | ✓ |
| Has two-dimensional Task | × | × | × | × | × | ✓ |
| Has Supervised Model Baseline | ✓ | × | × | ✓ | × | ✓ |
| Has Expert Model Baseline | × | × | × | ✓ | ✓ | ✓ |
| Has DNA Foundation Model Baseline | × | ✓ | ✓ | ✓ | ✓ | ✓ |

**Table 1:** Comparison of existing benchmarks (Genomic Benchmarks (Grešová et al., 2023), NT Benchmark (Dalla-Torre et al., 2023), GUE (Zhou et al., 2023), BEND (Marin et al., 2023) and LRB (Kao et al., 2024)) for DNA prediction tasks with DNALONGBENCH. While recent benchmarks have been proposed for genomics, only BEND and LRB address tasks involving relatively long-range dependencies. In contrast, DNALONGBENCH offers the most extensive range of long-range tasks, encompassing sequences up to 1 million base pairs. It also includes a greater variety of task types, longer input sequences, and evaluates the performance of three representative baseline models: supervised, expert, and DNA foundation models. The supervised baseline represents fully supervised models, such as lightweight CNNs, that do not involve pre-training.

## 2.2 LONG-RANGE DNA SEQUENCE MODELING

In the last decade, deep learning models for genomics have been dominated by CNNs (Consens et al., 2023). While CNNs excel at extracting local features, their limited local receptive field constrains the information flow between distant genomic elements (Avsec et al., 2021a). To address this limitation, researchers have introduced dilation convolution and skipped connections to CNN models. One such example is the Akita model (Fudenberg et al., 2020), which is designed to predict chromatin contact maps (i.e., chromatin folding) from DNA sequences up to around 1 million base pairs. Akita employs successive dilated convolutional layers and residual connections to enable information flow across long distances.

Unlike CNNs, which require many successive layers to capture long-range dependencies due to their local receptive field, transformers leverage the attention mechanism that allows each position in the sequence to directly attend to all other positions (Vaswani et al., 2017). However, transformers suffer from computational inefficiency on long sequences, as the attention mechanism scales quadratically with sequence length (Gu & Dao, 2023). This makes direct base-to-base attention across extremely long genomic sequences, spanning millions of base pairs, impractical (Gu & Dao, 2023; Schiff et al., 2024). To address this, hybrid models have been developed that combine convolutional layers for feature extraction with transformer modules. The Enformer model (Avsec et al., 2021a), for instance, integrates convolutional layers and transformers to predict epigenetic and transcriptional features across long DNA sequences up to 200k bases.

Recently, DNA foundation models have emerged as an active area of research (Zaheer et al., 2020; Ji et al., 2021; Dalla-Torre et al., 2023; Zhou et al., 2023; Nguyen et al., 2024b; Schiff et al., 2024). These models are pre-trained on large-scale DNA sequences and show potential across various downstream genomics tasks (Dalla-Torre et al., 2023; Zhou et al., 2023; Nguyen et al., 2024b). However, transformer-based DNA foundation models typically have relatively short context lengths (up to 4k tokens) due to the computational constraints of the self-attention mechanism (Dalla-Torre et al., 2023; Nguyen et al., 2024b). To solve this, researchers are proposing new transformer variants (Ding et al., 2023) and alternative architectures beyond transformers (Nguyen et al., 2024b; Gu & Dao, 2023; Schiff et al., 2024). For example, HyenaDNA (Nguyen et al., 2024b) is a non-transformer-based DNA foundation model that relies on implicit convolutions, allowing for long context lengths up to 1 million base pairs. It has demonstrated promising performance in long-range species classification tasks, even though the practical applications of this problem remain poorly defined. Caduceus (Schiff et al., 2024) is a bi-directional equivalent long-range DNA foundation model built on Mamba blocks (Gu & Dao, 2023). In this study, we selected HyenaDNA and Caduceus as part of the DNA foundation model baseline methods for evaluation in DNALONGBENCH, as they are specifically designed for long-range DNA prediction tasks.

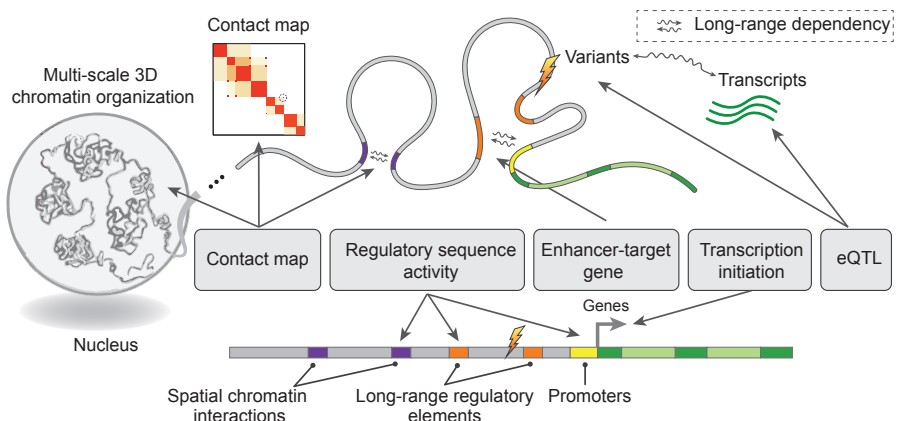

**Figure 1:** Illustration of the different categories of downstream tasks included in DNALONGBENCH.

| LR Tasks | LR Type | Input Length | Output Shape | # Samples | Metric |
|---|---|---|---|---|---|
| Enhancer-target Gene | Binary Classification | 450,000 | 1 | 2602 | AUROC |
| eQTL | Binary Classification | 450,000 | 1 | 31282 | AUROC |
| Contact Map | Binned (2048bp) 2D Regression | 1,048,576 | 99681 | 7840 | SCC&PCC |
| Regulatory Sequence Activity | Binned (128bp) 1D Regression | 196,608 | Human: (896, 5313) Mouse: (896, 1643) | Human: 38171 Mouse: 33521 | PCC |
| Transcription Initiation Signal | Nucleotide-wise 1D Regression | 100,000 | (100000, 10) | 100000* | PCC |

**Table 2:** Overview of the tasks included in DNALONGBENCH. 1D and 2D denote one-dimensional and two-dimensional, respectively. Nucleotide-wise tasks involve predicting a sequence of labels, each corresponding to individual nucleotides in the input. Sequence-wise tasks require classifying the entire input sequence. In binned tasks, multiple nucleotides are grouped into bins and share a common label. *: The data for this task consists of sequences sampled from whole genomes, with 100,000 being the number of samples used for training our baselines. AUROC: Area Under the Receiver Operating Characteristic curve. PCC: Pearson correlation coefficient. SCC: stratum adjusted correlation coefficient.

# 3 PROPOSED DATASET: DNALONGBENCH

It is important to select suitable long-range DNA prediction tasks for DNALONGBENCH to ensure diversity, comprehensiveness, and rigor. To achieve this, we established several criteria to guide our task selection process:

**Biological Significance:** Tasks should be realistic and biologically significant, each representing genomics problems important for understanding genome structure and function.

**Long-range Dependencies:** Tasks should require the modeling of long input contexts, spanning hundreds of kilobase (kb) pairs or more.

**Task Difficulty:** Tasks should pose significant challenges to current models.

**Task Diversity:** Tasks should be as diverse as possible, spanning various length scales and including different task types such as classification or regression. This diversity also extends to the task dimensionality (1D or 2D) and granularity (binned, nucleotide-wide, or sequence-wide).

As a result, we selected five long-range DNA prediction tasks, each covering various aspects of important regulatory elements and biological processes within a cell, as illustrated in **Figure** 1. An overview of our dataset is presented in **Table** 2. The input sequences for all tasks are provided in BED format, which lists the genome coordinates of the sequences. This format allows for flexible adjustment of the flanking context without reprocessing. The selected tasks are detailed in the following sections.

## 3.1 ENHANCER-TARGET GENE PREDICTION

**Definition:** In eukaryotic cells, enhancers play a key role in gene regulation by forming enhancer-promoter interactions that activate the transcription of target genes, even those located up to several mega basepairs away (Schoenfelder & Fraser, 2019). However, the detailed mechanism by which sequence information encodes enhancer-promoter interactions remains poorly understood. This task aims to predict true enhancer-promoter interactions from a list of putative candidates based on the DNA sequence.

**Biological Significance:** Although enhancers were initially believed to interact with target genes in an orientation- and distance-independent manner, recent findings suggest that the sequence and epigenomic signals within the window between enhancers and promoters may also contain important features for predicting enhancer-promoter interactions (Whalen et al., 2016). Predictive methods that incorporate the full sequence information between enhancers and promoters as input may not only improve the prediction performance but also help identify the sequence determinants driving these interactions.

**Data:** We collected experimentally verified enhancer-promoter interactions in K562 cells from Fulco et al. (2019), Gasperini et al. (2019) and Schraivogel et al. (2020). Using CRISPRi-based screening technique, the authors perturbed thousands of candidate sequences, quantified their effects on gene expression, and identified both positive and negative enhancer-promoter interactions. We filtered this data by retaining enhancer-promoter pair candidates within 450kb of the gene transcription start site (TSS). Genes with fewer than two positive pairs, two negative pairs, or five combined pairs were excluded. We then extracted the sequence between enhancers and promoters, extending 500bp upstream of the enhancer and 3kb downstream of the gene TSS. To remove bias from potential enhancers located within the interval between an enhancer and promoter pair, we masked the sequence of all intervening enhancers. Using a stratified sampling approach, the entire dataset was randomly split into training, validation, and test sets with an 8:1:1 ratio. A restriction was put in place to ensure that at least one positive and one negative pairs existed in both the training and validation sets.

**Evaluation:** We evaluated the performance of predictive models using AUROC. We compared models using sequence information alone with a task-specific expert model, activity-by-contact (ABC) model (Fulco et al., 2019), which incorporates DNase-seq, H3K27ac ChIP-seq data, and a Hi-C matrix to prioritize true enhancer-promoter interaction. It should be noted that the ABC model inherently have advantages over sequence-only models due to its more comprehensive input data types. The motivation here is mainly to compare sequence-only models and understand their limitations and strengths.

## 3.2 3D CHROMATIN CONTACT MAP PREDICTION

**Definition:** Contact map prediction refers to the 2D regression task of predicting the pairwise chromatin interactions between every pair of genomic loci within a given context window. These contact frequencies are expressed as 2D contact maps, which are derived from genomic mapping data such as Hi-C and Micro-C (Zhang et al., 2024).

**Biological Significance:** Chromosomes are folded in a well-organized manner within the cell nucleus, which affects various critical cellular functions, such as gene transcription and DNA replication (Misteli, 2007; Bonev & Cavalli, 2016). Developing prediction models that connect 1D DNA sequences and 2D contact map allows us to identify key sequence determinants of 3D chromatin folding, providing valuable insights into the underlying mechanisms of genome organization (Yang & Ma, 2022; Zhang et al., 2024).

**Data:** We used the processed data from Akita (Fudenberg et al., 2020), which includes chromatin interaction data from five cell lines: HFF, H1-hESC, GM12878, IMR-90, and HCT116. To increase the number of cell types, we also curated and processed additional Hi-C for four cell lines: HAP1, Hela, HepG2, and K562 using the same data processing steps used in the Akita model. Each input sequence, with a length of 1 million base pairs (Mbp), is divided into 512 genomic bins of 2kb resolution. For the final prediction, 32 genomic bins are cropped from each side, resulting in a contact map of 448×448. Since the contact map is symmetric, predictions are made only for the upper triangular region, with a diagonal offset of 2. The authors split the human genome into non-overlapping virtual contigs and randomly assigned them to training, validation, and testing sets

with an 8:1:1 ratio. The dataset contains 7,008 training sequences, 419 validation sequences, and 413 test sequences.

**Evaluation:** We used the Stratum-Adjusted Correlation Coefficient (SCC) and the Pearson Correlation Coefficient (PCC) to evaluate performance on the held-out test set.

## 3.3 REGULATORY SEQUENCE ACTIVITY PREDICTION

**Definition:** Cell type-specific regulatory activities are encoded by the compositions and interactions of functional DNA segments, such as promoters, enhancers, and insulators. The aim of this task is to predict thousands of epigenomic profiles on these DNA segments from DNA sequence alone up to 100kb. We set up this task by compiling human and mouse genomic tracks compiled in the Enformer paper (Avsec et al., 2021a). We formulated the task as a multitask regression problem that aims to predict epigenetic and transcriptional signals from long DNA sequences alone.

**Biological Significance:** Regulatory elements, such as promoters and enhancers, play crucial roles in controlling gene expression. These elements can regulate genes from distant locations across the genome. Predicting functional signals directly from DNA sequences over large genomic distances can help identify distal regulatory elements and uncover the key sequence features that enable them to control gene expression over long ranges.

**Data:** The dataset consists of experimentally determined regulatory activity signal tracks and corresponding DNA sequences from human and mouse genomes. The input DNA sequences are 196,608 bp, centered on the TSS of protein-coding genes. Each input sequence consists of a core region and flanking regions. The core sequence is 114,688 bp in length, corresponding to 896 bins with a resolution of 128 bp per bin. The target labels consist of 5,313 human tracks and 1,643 mouse tracks measuring epigenomic marks. The dataset contains 38,171 human sequences and 33,521 mouse sequences. For the human genome, the data is split into 34,021 training, 2,213 validation, and 1,937 test sequences. For the mouse genome, the dataset includes 29,295 training, 2,209 validation, and 2,017 test sequences.

**Evaluation:** We used Pearson correlation coefficient to evaluate model performance by comparing the predicted and target signal tracks. Specifically, Pearson correlation coefficient is computed for each sample using all positions and all tracks, and the mean is taken across all samples in the test set.

## 3.4 EXPRESSION QUANTITATIVE TRAIT LOCI (EQTL) PREDICTION

**Definition:** Expression quantitative trait loci (eQTL) are nucleotide variants that affect the expression of one or more genes. This task is derived from the dataset used in Enformer (Avsec et al., 2021a), where the goal is to predict whether a nucleotide variant can modulate the expression of a target gene using DNA sequence alone. Positive single-nucleotide polymorphisms (SNPs) are identified through a statistical fine-mapping tool Susie (Wang et al., 2020).

**Biological Significance:** Deep learning-based approaches to predicting gene expression from DNA sequences have gained increasing popularity. One practical application of these methods is the identification and interpretation of eQTLs, which is traditionally labor-intensive and time-consuming when relying on genome-wide association studies. This benchmark dataset provides an efficient way of evaluating eQTLs.

**Data:** The original datasets contain positive and matched negative variants across 48 tissues (Avsec et al., 2021a). For this study, we selected the top 9 tissues based on the number of variants. Within these tissues, we filtered eQTL-gene pairs by retaining eQTL candidate loci within 450kb of the gene TSS. Genes that have fewer than two positive pairs, two negative pairs, or five combined pairs were removed. We then extracted the sequence between variants and promoters, extending 3kb downstream of the gene TSS. To remove the bias caused by any putative eQTLs within the interval between an eQTL candidate and gene promoter pair, we masked the sequence of all variants within each variant-promoter pair. Using a stratified sampling approach, the dataset was randomly split into training, validation, and test sets with an 8:1:1 ratio. A restriction was put in place to make sure at least one positive and one negative pair exist in both the training and validation sets.

**Evaluation:** We evaluated the performance of predictive models using AUROC.

| Models | ETGP | CMP | | | | | |
|---|---|---|---|---|---|---|---|
| | K562 | HFF | H1hESC | GM12878 | IMR90 | HCT116 | Avg |
| Expert Model | **0.926** | **0.258** | **0.247** | **0.227** | **0.210** | **0.210** | **0.230** |
| CNN | 0.797 | 0.025 | 0.024 | 0.010 | 0.013 | 0.001 | 0.015 |
| HyenaDNA | 0.828 | 0.139 | 0.122 | 0.099 | 0.097 | 0.118 | 0.115 |
| Caduceus-Ph | 0.826 | 0.153 | 0.130 | 0.101 | 0.138 | 0.145 | 0.133 |
| Caduceus-PS | 0.821 | 0.142 | 0.123 | 0.097 | 0.132 | 0.139 | 0.127 |

**Table 3:** AUROC for enhancer-target gene prediction (ETGP) task and SCC scores for contact map prediction (CMP) task. K562, HFF, H1hESC, GM12878, IMR90, and HCT116 represent different human cell types. We highlight the highest scores in bold. Avg means the average score across different cell types. Note that Expert Model achieves the best performance on both ETGP and CMP tasks.

### 3.5 TRANSCRIPTION INITIATION SIGNAL PREDICTION

**Definition:** This task involves predicting the transcription initiation signal profile from DNA sequence. Specifically, it aims to predict transcription initiation signals on both strands for five experimental techniques: FANTOM CAGE, ENCODE CAGE, ENCODE RAMPAGE, GRO-cap, PRO-cap (Dudnyk et al., 2024). Unlike the regulatory sequence activity prediction task, which predicts sequence coverage at 128 bp genomic bins, this task requires predictions of transcription initiation signals at base pair resolution.

**Biological Significance:** Promoters are specialized DNA sequences at TSS of genes that support the assembly of the transcription machinery and transcription initiation (Haberle & Stark, 2018). Each promoter exhibits a unique profile of transcription initiation signals, which may reflect the mechanisms of transcription initiation. Solving the machine-learning task of predicting these profiles from promoter sequences would provide insights into sequence-based regulation of transcription initiation (Dudnyk et al., 2024). Previous studies have shown that regulatory elements can influence gene expression from far greater than tens of kb away (Avsec et al., 2021a). Using long sequences as input and improving the information flow between distal elements could enhance the predictive accuracy of transcription initiation signal prediction.

**Data:** We used processed labeled data from the Puffin paper (Dudnyk et al., 2024). The training set consists of 100kb intervals randomly sampled from all chromosomes, except for chromosomes 8, 9, and 10. Chromosome 10 is used for validation and chromosome 8,9 are for testing. In our study, we used 100,000 samples for training the baselines.

**Evaluation:** Following the Puffin paper (Dudnyk et al., 2024), predictions were generated for the entire test chromosomes (chr8 and chr9) using a sliding window step size of 50kb, with the center 50kb of each 100kb prediction being evaluated. Regions within 1kb of unknown bases or within 25kb of chromosome ends were excluded. Pearson's Correlation was used as the evaluation metric.

More details on data processing, licensing information, and the data link are provided in Appendix A, B, and C, respectively.

## 4 EXPERIMENTS

In this section, we conduct a comprehensive performance comparison by evaluating three distinct types of models, including lightweight convolutional neural network, existing expert models that has shown state-of-the-result results, and two types of very recent DNA foundation models, HyeynaDNA (Nguyen et al., 2024b) and Caduceus (Schiff et al., 2024) distinguished by their support to the reverse complement DNA during the training process.

### 4.1 REPRESENTATIVE MODELS

We explore the performance of the following three types of models:

| Models | RSAP | | | TISP | | | | | |
|---|---|---|---|---|---|---|---|---|---|
| | Human | Mouse | Avg | FC | EC | ER | GC | PC | Avg |
| Expert Model | **0.669** | **0.479** | **0.574** | **0.808** | **0.710** | **0.749** | **0.624** | **0.774** | **0.733** |
| CNN | 0.538 | 0.323 | 0.431 | 0.029 | 0.038 | 0.043 | 0.037 | 0.066 | 0.042 |
| HyenaDNA | 0.298 | 0.396 | 0.347 | 0.138 | 0.124 | 0.118 | 0.112 | 0.168 | 0.132 |
| Caduceus-Ph | 0.301 | 0.400 | 0.349 | 0.114 | 0.088 | 0.088 | 0.097 | 0.154 | 0.109 |
| Caduceus-PS | 0.301 | 0.387 | 0.344 | 0.113 | 0.088 | 0.090 | 0.102 | 0.156 | 0.108 |

**Table 4:** Pearson correlation scores for regulatory sequence activity prediction (RSAP) task and transcription initiation signal prediction (TISP) task. FANTOM CAGE (FC), ENCODE CAGE (EC), ENCODE RAMPAGE (ER), GRO-cap (GC), and PRO-cap (PC) denote the different experimental techniques utilized in the transcription initiation signal prediction task. Avg means average scores. We highlight the highest scores in bold. Note that the Expert Model outperforms both CNN and DNA foundation models by a large margin.

| Models | eQTLP | | | | | | | | | |
|---|---|---|---|---|---|---|---|---|---|---|
| | CCF | WB | Thyroid | SNSES | SSELL | MS | NT | AT | AS | Avg |
| Expert Model | **0.639** | **0.689** | **0.612** | **0.710** | **0.700** | **0.621** | **0.683** | **0.741** | **0.736** | **0.681** |
| CNN | 0.547 | 0.577 | 0.487 | 0.499 | 0.499 | 0.502 | 0.516 | 0.576 | 0.551 | 0.528 |
| HyenaDNA | 0.584 | 0.512 | 0.529 | 0.471 | 0.544 | 0.487 | 0.511 | 0.479 | 0.513 | 0.514 |
| Caduceus-Ph | 0.597 | 0.594 | 0.527 | 0.586 | 0.574 | 0.538 | 0.588 | 0.547 | 0.541 | 0.565 |
| Caduceus-PS | 0.549 | 0.542 | 0.547 | 0.529 | 0.541 | 0.523 | 0.552 | 0.536 | 0.519 | 0.537 |

**Table 5:** AUROC scores for expression quantitative trait loci prediction (eQTLP) task across different cell types. We abbreviate CCF for Cells_Cultured_fibroblasts, WB for Whole_Blood, SNSES for Skin_Not_Sun_Exposed_Suprapubic, SSELL for Skin_Sun_Exposed_Lower_leg, MS for Muscle Skeletal, NT for Nerve Tibial, AT for Artery_Tibial and AS for Adipose_Subcutaneous, i.e. the different cell types. The highest scores are highlighted in bold. Avg means the average score. Note that the Expert Model achieves the best performance across all cell types.

(1) **CNN**: We evaluate the performance of a lightweight convolutional neural network (LeCun et al., 2015), known for its simplicity and robust performance across various DNA-related tasks. Detailed model implementation for each task is provided in Appendix D.1.

(2) **Expert Model**: We assess the current state-of-the-art models for each specific long-range DNA prediction task, collectively referred to as the expert model. Specifically, we use:

- The Activity-by-Contact (ABC) model (Fulco et al., 2019) for the enhancer-promoter interaction prediction.
- The Enformer (Avsec et al., 2021a) for the eQTL prediction and regulatory sequence activity prediction.
- Akita (Fudenberg et al., 2020) for contact map prediction.
- Puffin-D (Dudnyk et al., 2024) for transcription initiation signal prediction.

More detailed information about each expert model is provided in Appendix D.2.

(3) **DNA Foundation Model**: We selected three long-range DNA foundation models – HyenaDNA (medium-450k) and Caduceus (Ph and PS) – as the DNA foundation models evaluated in this study. Due to the limited computing resources, we are not able to finetune HyenaDNA-large-1m and Evo (7B, (Nguyen et al., 2024a)). The detailed finetuning strategy for each task is provided in Appendix D.3.

### 4.2 BENCHMARKING RESULTS

The main results are reported in **Table** 3, **Table** 4 and **Table** 5. We also provide results on additional metrics in Appendix E.

**The Expert Model achieves the highest scores on all tasks.** Specifically, the Expert Model achieves an average score of 0.733 on the transcription initiation signal prediction task (TISP), significantly surpassing CNN's 0.042, HyenaDNA's 0.132, Caduceus-Ph's 0.109 and Caduceus-PS's 0.108. This

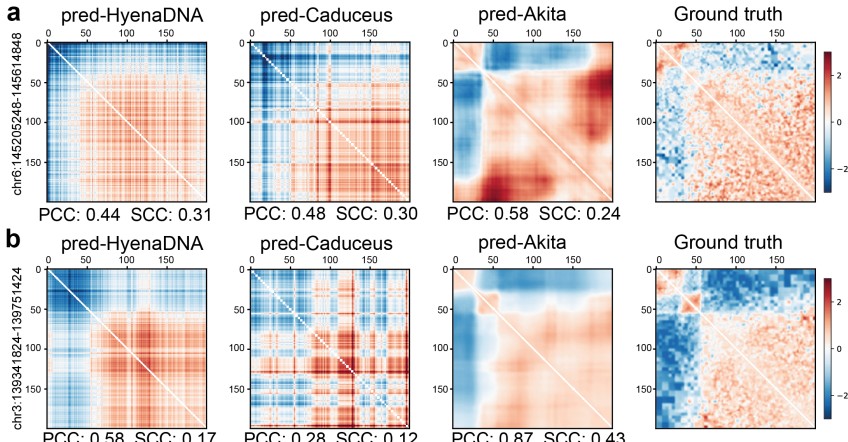

**Figure 2:** Comparisons of HyenaDNA, Caduceus (Ph), and the Expert Model (Akita) on the 2D contact map prediction task across 409,600 bp with a bin size of 2,048 bp. The columns show the contact map predicted by HyenaDNA, Caduceus, Akita model, and the ground truth contact map for two genomic regions: **(a)** chr6:145,205,248-145,614,848 and **(b)** chr3:139,341,824-139,751,424, respectively. Colors represent the intensity of contact frequency between paired loci. Pearson correlation coefficient (PCC) and stratum-adjusted correlation coefficient (SCC) metrics are shown under each contact map to indicate the prediction performance compared to the ground truth.

disparity may be attributed to the challenge posed by multi-channel regression on long DNA contexts, which makes the fine-tuning of DNA foundation models less stable and less capable of capturing sparse real-valued signals. In the remaining four tasks, the Expert Model still outperforms both CNN and DNA foundation models, although the difference is less pronounced. For example, Caduceus-Ph's performance on the contact map prediction task (CMP) is only slightly lower than the Expert Model and much better than CNN. Overall, these observations confirm the Expert Model's superior ability to capture long-range dependencies, a capability in which CNN falls short and DNA foundation models demonstrate good performance in certain tasks.

**The regulatory sequence activity prediction presents greater challenges.** In contrast to the other four tasks, where the Expert Model or DNA foundation models demonstrate decent performance, the regulatory sequence activity prediction task proves to be significantly more difficult. The highest average Pearson correlation score achieved in this task is 0.574 by the Expert Model (Enformer), indicating only a medium positive correlation. This result highlights the challenge of capturing long-range dependencies in regulatory sequence activity prediction and further validates the varying levels of task difficulty in our proposed DNALONGBENCH.

## 5  ANALYSIS: DIVING DEEP INTO DNALONGBENCH

In this section, we provide further analysis to gain an insight into how long-range dependencies function in our proposed DNALONGBENCH.

### 5.1  CASE STUDY: CAN LONG-RANGE DEPENDENCY BE CAPTURED?

To intuitively demonstrate that extensive long-range dependencies exist across millions of base pairs and can be captured by machine learning methods, we present two examples in **Fig.** 2. **Fig.** A1 contains the three more examples. Specifically, in **Fig.** 2 (a) and (b), we visualize the contact maps sequentially predicted by HyenaDNA, Caduceus-Ph and the Expert Model (Akita), alongside the ground truth contact maps on the right, for two genomic regions with spanning around 400kb. Based on these contact maps, we observe the presence of large-scale domains (e.g., blocks on the contact map) and long-range interactions (e.g., dots on the contact map) spanning over 300kb. Notably, the contact matrices predicted by Akita align more closely with the ground truth, confirming its superior ability to capture long-range interactions. In contrast, the DNA foundation models demonstrate limited capacity for predicting domain structures. This is particularly evident in **Fig.** 2 (b), where

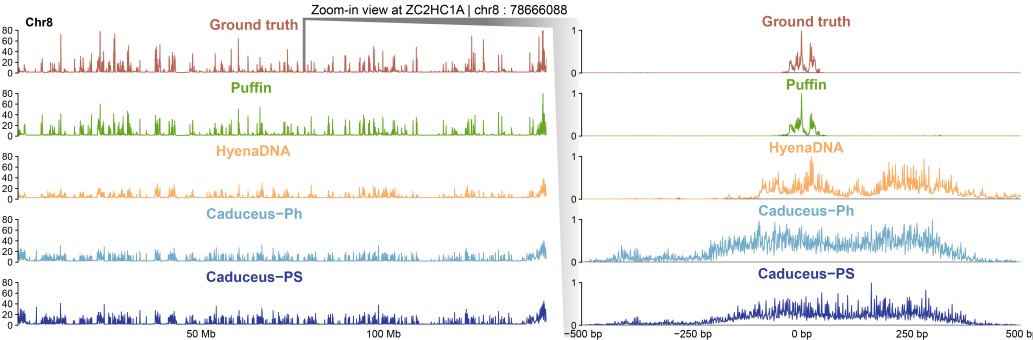

**Figure 3:** Comparisons of HyenaDNA, Caduceus_Ph,Caduceus_PS, and Expert Model (Puffin-D) on the transcription initiation signal prediction task of chromosome 8. The genomic track on the left shows the ground truth signals (top) and the predictions by Puffin-D, HyenaDNA, and two Caduceus models. The x-axis represents genomic coordinates, while the y-axis indicates signal density. A zoomed-in view of a 1,000 bp region centered at the transcription start site of the gene *ZC2HC1A* is shown on the right.

only Akita accurately predicts the three blocks. These examples demonstrate that DNALONGBENCH is valuable for evaluating models that capture long-range genome structure and function. We believe this will inspire and inform the future development of DNA foundation models.

### 5.2 BASE PAIR-RESOLUTION PREDICTION OF TRANSCRIPTION INITIATION SIGNAL

We visualized the transcription initiation signals predicted by different models for one of the test chromosomes, chr8 (**Fig.** 3). The predictions of the expert model, Puffin-D, closely align with the ground truth, accurately capturing the peaks in transcription initiation signal intensity across both large and small genomic regions. On the other hand, DNA foundation models tend to underpredict signal intensities or miss certain peaks. Notably, in the zoomed-in view on the right side of the figure, Puffin continues to align well with the ground truth, demonstrating strong performance even at a high resolution. In contrast, the DNA foundation models show broader, less precise signals. In summary, long-range base pair-resolution regression task remains challenging for DNA foundation models.

## 6 CONCLUSION

In this paper, we introduce DNALONGBENCH, a benchmark suite consisting of five important genomics tasks that involve long-range dependencies: enhancer-target gene interaction, eQTL, 3D genome organization, regulatory sequence activity, and transcription initiation signal. We evaluated three baseline methods: a task-specific expert model, a fully supervised CNN-based model, and three fine-tuned DNA foundation models, HyenaDNA, Caduceus-Ph and Caduceus-PS. The benchmarking results consistently showed that expert models achieved the highest scores across all tasks. Furthermore, our analysis demonstrated that long-range dependencies could be captured across hundreds of thousands of base pairs, illustrating the importance of considering context length in downstream performance. However, it is also clear that current DNA foundation models are not as effective in capturing long-range dependencies compared to expert models. Nevertheless, we hope that DNALONGBENCH will serve as a useful resource, facilitating comprehensive comparisons and rigorous evaluations of emerging DNA sequence-based deep learning models that account for long-range dependencies.

One limitation of this paper is that we did not evaluate transformer-based DNA foundation models, such as DNABERT-1, DNABERT-2, and Nucleotide Transformer. This is primarily due to the computational challenges posed by training them on long-range tasks, as the quadratic cost of the self-attention mechanism often renders such tasks infeasible. Exploring strategies to extend the context length of these models and effectively fine-tune them for long-range tasks represents an important direction for future research, although it is beyond the scope of this study.

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

# A APPENDIX

# A DATASET DOCUMENTATION AND INTENDED USES

We document datasets of this study following the Datasheets for Datasets framework (Gebru et al., 2021), discussing Motivation, Composition, Collection process, Preprocessing and Uses.

## A.1 ENHANCER-TARGET GENE PREDICTION

**Motivation** This dataset is created to explore the predictive power of using DNA sequence alone to predict the correct target genes of an enhancer. The dataset can be used for various DNA large language models to test their capabilities in capturing the long-range dependency of DNA regulatory elements.

**Composition** This dataset contains multiple DNA sequences of nucleotide letters A, C, G, T, and N, where N is used to indicate an unknown nucleotide or as a padding sequence. These sequences represent the DNA sequence between enhancers and their putative target genes. The genes are always located on the 5' end of the DNA sequence and the enhancers are located on the 3' end of the DNA sequence. The length of the DNA sequence is 450,000bp, and we use 'N' as the padding nucleotide on the 3' end when the distance between the enhancer and the target gene is smaller than 450,000bp. Each sequence is associated with a binary label indicating whether the enhancer can regulate the target gene or not. This dataset contains a total of 2602 samples, partitioned into 2066 training sequences, 266 validation sequences, and 270 test sequences. We split the dataset using stratified sampling with the restriction that each gene must have at least one positive and negative enhancer-gene pair in both the training and validation sets. Overall, this dataset contains tested enhancers for 24 genes.

**Collection Process** The original enhancer-gene pairs were reported in Fulco et al. (2019), which used the CRISPRi-FlowFISH based method to experimentally determine whether an enhancer candidate can regulate the target gene. For each gene, dozens of enhancer candidates were tested for their abilities to regulate gene expression. Enhancers that can significantly change the expression of target genes with a False Discovery Rate (FDR) of less than 0.05 are considered positive enhancer-gene pairs. The remaining enhancers are considered negative enhancers for that specific gene.

**Preprocessing** The following processing steps were implemented to create the benchmark dataset. We first collected and curated CRISPRi-based screening data from Fulco et al. (2019), Gasperini et al. (2019) and Schraivogel et al. (2020). For each enhancer-gene pair, we extracted the DNA sequence between the enhancer and its target gene using the human hg19 reference genome assembly (liftover to hg19 if in a different genome assembly) downloaded from the UCSC Genome browser We excluded enhancers that are located more than 450,000 base pairs (bp) away from the gene's transcription start site. The sequence surrounding the enhancers and the transcription start site may also contribute to gene regulation. Therefore, we also included an additional 3000bp of DNA sequence downstream of the transcription start site and 500bp upstream of the enhancer candidate. Additionally, we masked the DNA sequence of any other enhancers tested in the paper between each enhancer and its target gene, to remove any potential biases caused by these additional enhancers. We add the letter 'N' to the 3' end of the sequence to make all the sequences the same length of 450,000bp.

**Uses** The authors of Fulco et al. (2019) have used this dataset to test the performance of their proposed activity-by-contact (ABC) model.

## A.2 CONTACT MAP PREDICTION

**Motivation** The Contact Map Prediction dataset is created by the authors of Akita (Fudenberg et al., 2020) to explore the connection between DNA sequences and 3D genome structure. Given a DNA sequence, a model predicts the interactions between each pair of genomic bins within the sequence.

**Composition** The Contact Map Prediction dataset contains human genome sequences and chromatin pairwise contact maps. Each sample consists of an input DNA sequence with a length of 1,048,576 bp, divided into 2,048-bp genomic regions, which results in 512 bins per sequence. Additionally, the dataset includes the output pairwise interaction frequencies for the central 448 bins, represented as a $448 \times 448$ 2D matrix. Since the contact map is symmetric, only the upper triangular region

of the contact map is used for prediction, with a diagonal offset of 2. Thus, the final output is a vector of length 99,681. The dataset contains 7,840 sequences in total, partitioned into 7,008 training sequences, 419 validation sequences, and 413 test sequences.

**Collection Process** The human genome sequence is collected from the human reference genome assembly GRCh38. The contact maps are obtained from the publicly available Hi-C and Micro-C data. There are nine cell types included in this dataset. The first five cell lines are the same as the training data of the Akita model, including HFF and H1-hESC from (Krietenstein et al., 2020), GM12878 and IMR-90 from (Rao et al., 2014), and HCT116 from (Rao et al., 2017). The data for the other four cell lines were collected from the 4DN data portal. These data were processed in the same way presented in the Akita. The raw interaction pairs were binned into 2048 bp bins. Functions in cooltools were used to process the binned contact map through normalization, adaptively coarse-grain smooth, and linearly interpolation of missing bins, and convolve with a small 2D Gaussian filter (sigma=1, width=5).

**Preprocessing** The following pre-processing steps are applied to enhance the quality of the contact maps: adaptive coarse-graining, normalization for the distance-dependent decreases, log transformation, clipping the values to the range of (-2, 2), linear interpolation of missing values, and convolution with a small 2D Gaussian filter.

**Uses** The dataset has been used by Akita (Fudenberg et al., 2020) to predict the chromatin interactions from DNA sequences.

### A.3 REGULATORY SEQUENCE ACTIVITY PREDICTION

**Motivation** The Regulatory Sequence Activity Prediction dataset aims to study the effects of DNA sequences on various epigenetic and transcriptional signals. With a model trained on the dataset, we can infer the regulatory impact of noncoding DNA on gene expression.

**Composition** The Regulatory Sequence Activity Prediction dataset contains DNA sequences with a length of 196,608 bp as input data, including 38,171 human sequences and 33,521 mouse sequences. For one sequence, the prediction targets are multiple genome-wide tracks, with 5,313 tracks for human and 1,643 tracks for mouse. These tracks are measured using four types of technologies: ChIP-seq, DNase-seq, ATAC-seq, and CAGE.

**Collection Process** The dataset is created by the authors of Enformer (Avsec et al., 2021a). They collect the dataset based on the Basenji2 dataset (Kelley, 2020) by extending the length of input DNA sequences from 131,072 bp to 196,608 bp.

**Preprocessing** We use the same training, validation, and test datasets as provided in the Enformer paper. No additional preprocessing steps here.

**Uses** This dataset has been used to train the Enformer model (Avsec et al., 2021a).

### A.4 EQTL PREDICTION

**Motivation** This dataset is created to explore the predictive power of using DNA sequence alone to identify expression quantitative trait loci (eQTL) from germline mutations.

**Composition** This dataset contains nine sub-datasets representing putative eQTL from 9 human tissues. Each sample of the dataset consists of a pair of DNA sequences with a length of 450,000bp from the mutation to the transcription start site of the putative target gene, one representing the reference sequence and another representing the alternative sequence. The only difference between the reference sequence and alternative sequence appears on the variants. The gene is always located on the 5' end of the sequence and the mutation is always located on the 3' end of the sequence. We use 'N' as the padding nucleotide on the 3' side when the distance between the mutation and target gene is smaller than 450,000bp. Each sample is also associated with a binary label indicating whether the mutation is eQTL or not. The number of samples varies from 2181 to 4919 across 10 tissues. The total number of samples is 31,282.

**Collection Process** The original dataset is created by the authors of Enformer (Avsec et al., 2021a). The authors collected the statistical fine-mapping variants dataset from GTEx v8 using the SuSiE method (Wang et al., 2020).

**Preprocessing** We picked the top 9 tissues from the original 48 tissues according to the number of variants. In each tissue, we filtered variant-gene pairs by retaining variants located within 450,000bp of the gene transcription state site. We then extracted the sequence between variants and genes after extending 3kb downstream of the gene TSS. To remove the bias caused by any putative eQTLs within the interval between an eQTL candidate and promoter pair, we further masked the sequence of all variants within each variant-promoter pair. We followed the authors of Enformer to identify positive and negative variants. Positive variants are variants with posterior inclusion probability (PIP) > 0.9. Negative variants are selected by matching to each positive variant from the set with PIP < 0.01 and the absolute value of Z-score > 4 tested for the same gene, or from the set with PIP < 0.01 and the absolute of Z-score > 6 genome-wide. We used stratified sampling to split the dataset into training, validation, and testing with a ratio 80:10:10 with the restriction that each gene must have at least two positive and two negative variants in both training and validation sets.

**Uses** This dataset has been used to test the performance of Enformer (Avsec et al., 2021a) in predicting the effects of variants in gene expression.

### A.5 TRANSCRIPTION INITIATION SIGNAL PREDICTION

**Motivation** The Transcription Initiation Signal Prediction dataset is to investigate how DNA sequences determine the transcription initiation process. The task is to predict transcription initiation signals at base-pair resolution from DNA sequences.

**Composition** The Transcription Initiation Signal Prediction dataset contains human genome sequences and transcription initiation signals generated by five experimental techniques, including two variants of CAGE from the FANTOM and ENCODE projects, RAMPAGE, GRO-cap, and PRO-cap. Each sample is composed of an input DNA sequence with a length of 100,000 and a multi-task label of 10 for each base, resulting in a final output shape of (100,000, 10). The 10 signals are from five experimental techniques for both the forward and reverse strands. The training set consists of 100,000 intervals randomly sampled from all chromosomes except for chromosomes 8, 9, and 10. Chromosome 10 is used for validation, and chromosomes 8 and 9 are used for testing. This dataset is essentially a transcription initiation map at base-pair resolution for the whole genome.

**Collection Process** The dataset is created by the authors of Puffin (Dudnyk et al., 2024). The human genome sequence is collected from the human reference genome assembly GRCh38. The transcription initiation signals comprise five experimental techniques, including two variants of CAGE (Shiraki et al., 2003) from the FANTOM (Consortium et al., 2014) and ENCODE (Consortium et al., 2012) project, RAMPAGE (Moore et al., 2022), GRO-cap (Core et al., 2014), and PRO-cap (Kwak et al., 2013).

**Preprocessing** We used the same random seed that the Puffin-D model used to sample the sequences from the training chromosomes. We stopped when we obtained 100,000 samples for training the baselines.

**Uses** This dataset has been used to train the Puffin model (Dudnyk et al., 2024).

## B LICENSE

DNALONGBENCH is licensed under CC BY 4.0. A copy of the license is provided with the dataset. Citing the original sources is required when using the data provided with DNALONGBENCH. The authors bear all responsibility in case of violation of rights.

## C DATA AVAILABILITY

Datasets included in DNALONGBENCH are available at:

- Regulatory Sequence Activity Prediction Data: https://dataverse.harvard.edu/privateurl.xhtml?token=4c6b250c-26fc-412a-b3e1-bc15f1332f0c

- Transcription Initiation Signal Prediction: https://dataverse.harvard.edu/privateurl.xhtml?token=9810103a-b8b8-4a4d-95c4-b26b6e153446

- Enhancer-Target Gene Prediction: https://dataverse.harvard.edu/privateurl.xhtml?token=c238c0dd-528f-4d04-a3c8-0ff1eee1d651

- 3D Chromatin Contact Map Prediction: https://dataverse.harvard.edu/privateurl.xhtml?token=a990b515-d76e-4b63-ba74-5c78c469ae53

- Expression Quantitative Trait Loci (eQTL) Prediction: https://dataverse.harvard.edu/privateurl.xhtml?token=93d446a5-9c75-44bf-be1c-7622563c48d0

## D  IMPLEMENTATION DETAILS

All experiments were performed on single GPUs of A6000 on clusters. We implemented all models using PyTorch (Paszke et al., 2017).

### D.1  CNN IMPLEMENTATION DETAILS

**Enhancer-Target Gene Prediction** The CNN model is trained with cross-entropy loss with a learning rate of 0.005. The batch size is set to 5 and the model is optimized using Adam. The architecture of the CNN consists of three Conv1D layers with 128, 64, and 32 filters respectively, followed by a fully connected layer to predict the class probabilities. Max pooling is added before the fully connected layer. The kernel sides for the convolutional layered are all 3 with padding of 1.

**Contact Map Prediction** The CNN model is trained using Adam with a learning rate of 0.005 for 30 epochs with a batch size of 16. The CNN model is specially designed for this 2D regression task. The architecture is composed of a 1D convolutional tower, a bottleneck layer, a 2D convolutional tower, a cropping layer, and a final transformation layer. The 1D convolutional tower consists of three Conv1D layers. A bottleneck layer is applied to the output of the convolutional tower, consisting of a 1D convolutional layer with 64 filters and a kernel size of 1. To convert the 1D sequence into a 2D map, the output from the bottleneck layer is unsqueezed and repeated to create a tensor of shape `[batch size, 64, 512, 512]`. A positional encoding is then concatenated to this tensor, resulting in a tensor of shape `[batch size, 65, 512, 512]`. This tensor is passed through three 2D convolutional layers. The output from the 2D convolutional tower is then cropped to remove 32 bins from each side, resulting in a tensor of shape `[batch size, 64, 448, 448]`. Following this, the upper triangular portion of the 2D map with a diagonal offset of 2 is extracted, resulting in a tensor of shape `[batch size, 64, 99681]`. The final transformation layer is a 1D convolutional layer that reduces the feature dimension to 1, resulting in the final output tensor of shape `[batch size, 1, 99681]`.

**Regulatory Sequence Activity Prediction** The CNN model is trained with Possion loss with a learning rate of 0.001. The batch size is set to 16 and the model is optimized using Adam. The architecture of the CNN consists of five layers, with convolutions of 16, 64, 256, 1024, and 5313 filters respectively for the human organism, and 16, 64, 256, 1024, and 1643 filters respectively for the mouse organism. The kernel sizes for the convolutional layers are set to 25, 15, 15, 15, and 1, respectively. Batch normalization and max pooling are added between each two layers. Specifically, adaptive max pooling is employed after the fourth layer to ensure the output sequence length matches the target length of 896.

**eQTL Prediction** The CNN model is trained with cross-entropy loss with a learning rate of 0.005. The batch size is set to 5 and the model is optimized using Adam. The architecture of the CNN consists of three Conv1D layers with 128, 64, and 32 filters respectively, followed by a fully connected layer to predict the class probabilities. Max pooling is added before the fully connected layer. The kernel sides for the convolutional layered are all 3 with padding of 1.

**Transcription Initiation Signal Prediction.** The CNN model is trained using Adam with a learning rate of 0.005 and a batch size of 16. The architecture of the CNN consists of three layers, with convolutions of 16, 32, and 64 filters, respectively. Mean Squared Error (MSE) is used as the loss function.

## D.2 EXPERT MODEL DETAILS

**Enhancer-Target Gene Prediction** We applied the actvity-by-contact (ABC) model to predict whether an enhancer candidate is a true enhancer or not. Specifically. The ABC score of each enhancer of its putative target genes is provided by Fulco et al. (2019). We then used these ABC scores to calculate the AUROC on the testing set.

**Contact Map Prediction** We used the Akita model (Fudenberg et al., 2020) as the expert model. Akita consists of two main component, a trunk that learns the 1D representations of the DNA sequences and a head that converts the 1D sequence representations into 2D contact map predictions. The trunk is composed of 11 convolution blocks followed by 8 dilated residual 1D convolutions with geometrically increasing dilation rate. The convolution blocks iteratively perform 1D convolution with 96 filters of width five, batch normalization, ReLU, and width two maximum pooling. A bottleneck width one covolution with 64 filters is added to the end of the trunk, resulting in an output of shape `[512 bins, 64 filters]`. The head first converts the 1D sequence representations with shape `[512, 64]` to 2D maps with shape `[512, 512, 64]` by averaging the representation between every pair of genomic bins $i$ and $j$. Additionally, the distance between the genomic bins $|i - j|$ is included as an extra positional feature. 6 blocks of dilated 2D convolutions, with geometrically increasing dilation rate, is applied to learn the 2D contact map. At the end of each convolution block, resymmetrization is performed by summing the contact map with its transpose and dividing by two. Finally, linear transformation is utilized to simultaneously predict the contact maps for all five cell types. We evaluated the trained Akita model instead of training from scratch.

**eQTL Prediction** Following the Enformer paper, we first use the pretrained Enformer model (Avsec et al., 2021a) to obtain the features of reference and alternative sequences. The variant can be represented as the prediction difference vector by subtracting these two features and summing the differences across the sequence. We then employ a random forest classifier to infer whether the variant is positive or negative. The random forest model is implemented by scikit-learn with 100 trees, and the maximum number of features considered when looking for the best split is set to $log_2$ of the total number of features.

**Transcription Initiation Signal Prediction.** We used the Puffin-D model (Dudnyk et al., 2024) as the expert model for this task. Puffin-D is a specially designed CNN-based model containing two upward and downward passes with residual connections. The architecture of Puffin-D consists of two upward blocks, two downward blocks, and one output block. The upward blocks are composed of strided convolutional layers followed by batch normalization. The downward blocks are composed of upsampling, strided convolutional layers, and batch normalization. The final output block consists of 1D convolutional layers with a kernel size of 1, batch normalization, ReLU activation, and Softplus activation. Similar to U-Net (Ronneberger et al., 2015), residual connections are implemented between corresponding levels of the upward and downward passes.

**Regulatory Sequence Activity Prediction** We evaluate the performance of the pre-trained Enformer model[1] instead of training the Enformer model (Avsec et al., 2021a) from scratch.

## D.3 DNA FOUNDATION MODEL FINETUNING DETAILS

**Enhancer-Target Gene Prediction.** First, we calculate the feature vector by averaging the hidden representations across the entire DNA sequence. Next, we use this extracted feature vector to perform a binary classification task. During this process, we fine-tune all model parameters for optimal performance.

**Contact Map Prediction.** First, we calculate feature vectors for each base pair bin, with a bin size of 2048, resulting in L/2048 bins, where L is the original DNA sequence length. We then calculate the contact map score by utilizing a two-layer multi-layer perceptron (MLP). This MLP takes concatenated feature vectors from two bins as input, resulting in a contact map matrix of dimensions [L/2048, L/2048]. We then extract the upper triangular part of this matrix and apply mean square error to train the model. All parameters are fine-tuned during this process.

**Regulatory Sequence Activity Prediction.** We follow the standard split provided by Enformer (Avsec et al., 2021a) and train two separate models for human and mouse. We use 128bp as

---

[1]https://huggingface.co/EleutherAI/enformer-official-rough

| Models | CMP | | | | | |
| --- | --- | --- | --- | --- | --- | --- |
| | HFF | H1hESC | GM12878 | IMR90 | HCT116 | Avg |
| Expert Model | **0.633** | **0.653** | **0.581** | 0.595 | 0.553 | **0.603** |
| CNN | 0.098 | 0.074 | 0.082 | 0.077 | 0.028 | 0.072 |
| HyenaDNA | 0.520 | 0.517 | 0.539 | **0.635** | **0.568** | 0.556 |
| Caduceus-Ph | 0.539 | 0.524 | 0.418 | 0.473 | 0.536 | 0.498 |
| Caduceus-PS | 0.528 | 0.454 | 0.391 | 0.424 | 0.508 | 0.461 |

**Table A1:** Pearson correlation scores for contact map prediction (CMP) task. K562, HFF, H1hESC, GM12878, IMR90, and HCT116 represent different human cell types. We highlight the highest scores in bold. Avg means the average score across different cell types.

| Models | HFF | H1hESC | GM12878 | IMR90 | HCT116 | Avg |
| --- | --- | --- | --- | --- | --- | --- |
| Caduceus-Ph-409600bps | 0.153 | 0.130 | 0.101 | 0.138 | 0.145 | **0.133** |
| Caduceus-Ph-307200bps | 0.066 | 0.076 | 0.051 | 0.054 | 0.149 | 0.079 |
| Caduceus-Ph-204800bps | 0.082 | 0.090 | 0.047 | 0.053 | 0.146 | 0.083 |

**Table A2:** Ablation study on context length for contact map prediction task.

a bin, resulting in 896 bins in total. We formulate it as a multi-regression task, where the output is [896,5313] for the human organism and [896,1643] for the mouse organism. We use Poisson loss and the batch size is set to 32. The maximum training step is 30k.

**eQTL Prediction.** The original dataset consists of triples in the format <original sequence, variant sequence, binary label>, where the binary label indicates whether the variant effect is positive or negative compared to the original sequence. To accomplish this task, we first average the hidden representations from the last layer for both the original and variant sequences. We then concatenate the two averaged feature vectors and apply a binary classification layer to predict if the variant sequence is positive or not. All parameters are fine-tuned during this process.

**Transcription Initiation Signal Prediction.** We formulate this task as a 10-channel regression task and use pseudo-Poisson KL divergence as the loss function. We add one linear layer on top of the model and predict logits of 10 signals at each position. We follow the Puffin (Dudnyk et al., 2024) to randomly sample 100k positions from the genome sequence during training. The batch size is 16 and the maximum training step is 25k.

## E  ADDITIONAL METRICS

To evaluate model performance in the Contact Map Prediction task, we used two key metrics: PCC (Pearson Correlation Coefficient) and SCC (Stratum Adjusted Correlation Coefficient). While it is common practice to use PCC when comparing two contact matrices, PCC does not account for domain structures or distance dependence, which are two unique characteristics of contact maps. To address these limitations, SCC was introduced by Yang et al. (2017), enabling more fine-grained differentiation between contact maps. Therefore, we included both metrics in our evaluations. **Table** A1 presents the benchmark results using PCC as the evaluation metric. **Fig.** A1 shows more examples of the comparison of different predictive models with the ground truth.

## F  ABLATION STUDY ON CONTEXT LENGTH

To validate that long-range context denpendencies are truly helpful for the tasks to good performance, we evaluate them under different lengths of the input context.

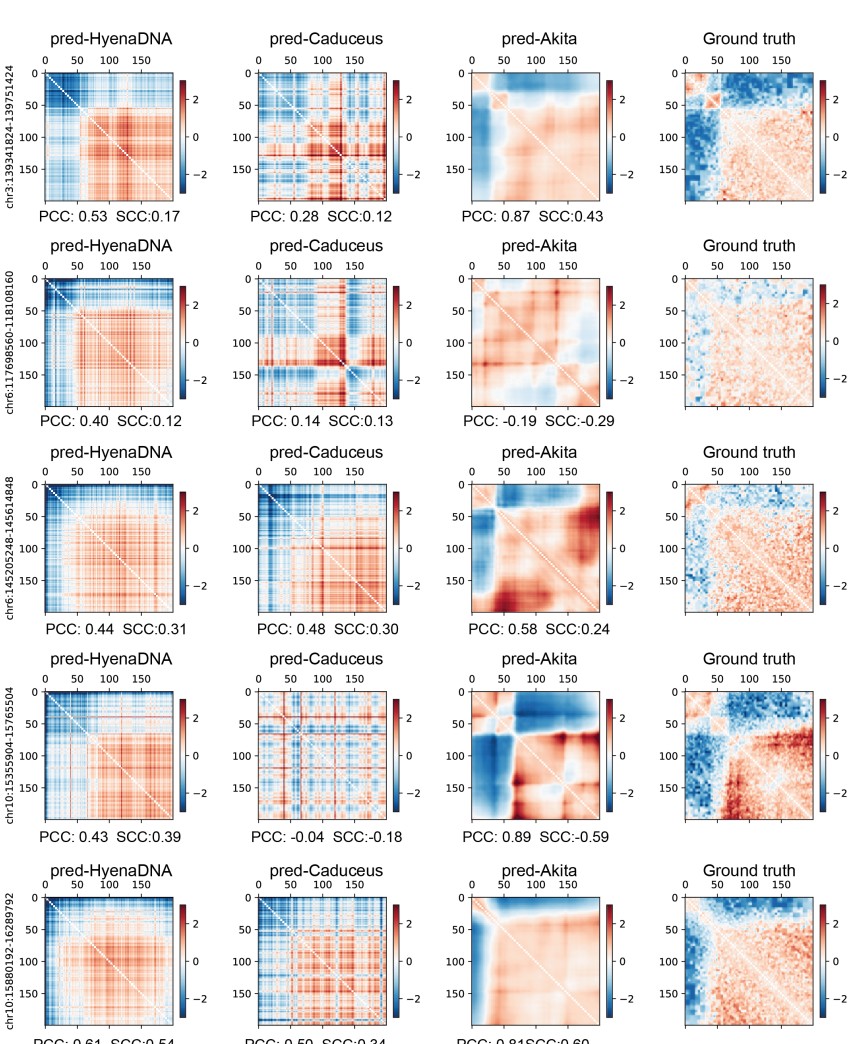

**Figure A1:** Comparisons of HyenaDNA, Caduceus (Ph), and the Expert Model (Akita) on the 2D contact map prediction task across 409,600 bp with a bin size of 2,048 bp. The columns show the contact map predicted by HyenaDNA, Caduceus, Akita model, and the ground truth contact map for five different genomic regions (rows) Colors represent the intensity of contact frequency between paired loci. Pearson correlation coefficient (PCC) and stratum-adjusted correlation coefficient (SCC) metrics are shown under each contact map to indicate the prediction performance compared to the ground truth.

| Models | RSAP | | |
| --- | --- | --- | --- |
| | Human | Mouse | Avg |
| HyenaDNA-196k | 0.298 | 0.396 | 0.347 |
| HyenaDNA-131k | 0.268 | 0.393 | 0.331 |

**Table A3:** Ablation study on context length for regulatory sequence activity prediction task.

| Models | ETGP |
| --- | --- |
| Caduceus-Ph | **0.826** |
| Caduceus-Ph w/ shuffled context | 0.418 |

**Table A4:** Ablation study on context length for enhancer-target gene prediction task.

| Models | eQTLP | | | | | | | | | |
| --- | --- | --- | --- | --- | --- | --- | --- | --- | --- | --- |
| | CCF | WB | Thyroid | SNSES | SSELL | MS | NT | AT | AS | Avg |
| Caduceus-Ph | 0.597 | 0.594 | 0.527 | 0.586 | 0.574 | 0.538 | 0.588 | 0.547 | 0.541 | **0.565** |
| w/ shuffled context | 0.519 | 0.515 | 0.507 | 0.533 | 0.539 | 0.521 | 0.542 | 0.507 | 0.504 | 0.521 |

**Table A5:** Ablation study on context length for eQTL prediction task.

| Models | TISP | | | | | |
| --- | --- | --- | --- | --- | --- | --- |
| | FC | EC | ER | GC | PC | Avg |
| Puffin-D | 0.593 | 0.331 | 0.294 | 0.249 | 0.394 | **0.372** |
| Puffin-D w/ shorter context | 0.530 | 0.283 | 0.450 | 0.212 | 0.331 | 0.361 |

**Table A6:** Ablation study on context length for TSIP task On the validation chromosome 10.

| Models | TISP | | | | | |
| --- | --- | --- | --- | --- | --- | --- |
| | FC | EC | ER | GC | PC | Avg |
| Puffin-D | 0.808 | 0.710 | 0.749 | 0.624 | 0.774 | **0.733** |
| Puffin-D w/ shorter context | 0.781 | 0.699 | 0.715 | 0.604 | 0.702 | 0.700 |

**Table A7:** Ablation study on context length for TSIP task On the validation chromosome 8-9.

### F.1 CONTACT MAP PREDICTION

We use Caduceus-Ph as an example here, and evaluate the model's performance with varying input sizes of 409600, 307200, and 204800 bps, corresponding to 200, 150, 100 bins, correspondingly. The evaluation metric utilized was stratum-adjusted correlation coefficient (SCC). The results are shown in Table A2. Our results show a clear trend as the context length decreased, the prediction performance also declined.

### F.2 REGULATORY SEQUENCE ACTIVITY PREDICTION

For the regulatory sequence activity prediction task, taking HyenaDNA as an example, we evaluated this task with two different lengths, 196k and 131k bps. The results are provided in Table A3. Our results show a clear trend as the context length decreased, the prediction performance also declined.

### F.3 ENHANCER-TARGET GENE PREDICTION

| Models | ETGP |
|---|---|
| Expert Model (ABC) | 0.896 |
| Expert Model (gABC) | 0.890 |
| CNN | 0.711 |
| HyenaDNA | 0.752 |
| Caduceus-Ph | **0.906** |
| Caduceus-ps | 0.895 |

**Table A8:** ETGP results on data curated from (Schraivogel et al., 2020)

| Models | ETGP |
|---|---|
| Expert Model (ABC) | 0.824 |
| Expert Model (gAB) | **0.831** |
| CNN | 0.730 |
| HyenaDNA | 0.781 |
| Caduceus-Ph | 0.788 |
| Caduceus-ps | 0.793 |

**Table A9:** ETGP results on data curated from (Gasperini et al., 2019)

| Cell Type | Accession Number | Data Source |
|---|---|---|
| HAP1 | 4DNFIWGGYEW2 | 4DN |
| Hela | 4DNFI65WJKMT | 4DN |
| HepG2 | 4DNFIQ4G74OW | 4DN |
| K562 | 4DNFI2R1W3YW | 4DN |

**Table A10:** Accession number and data source of the additional dataset curated for the contact map prediction task

For the enhancer-target gene prediction task, rather than simply reducing the input size, we preserved the enhancer sequences and those near gene promoter regions due to their critical role in interactions. To investigate how context length affects outcomes, we randomly shuffled the central portion of the input sequence with the length of 50% of the original input length. Taking Caduceus-Ph as an example, we observe a notable decline in performance when this central shuffling occurred (Table A4). These findings suggest that the DNA sequence lying between enhancers and gene promoters is also vital in predicting their interactions, consistent with previous observations (Whalen et al., 2016).

### F.4    EXPRESSION QUANTITATIVE TRAIT LOCI (eQTL) PREDICTION

For the eQTLP task, taking Caduceus-Ph as an example, we observe a similar tendency to ETGP (Table A5). When we performed central shuffling, the performance declined. It indicates that long sequence context is beneficial for these tasks to achieve better performance.

### F.5    TRANSCRIPTION INITIATION SIGNAL PREDICTION

For the TISP task, using the expert model Puffin-D as an example, we evaluated performance before and after using a short effective context on holdout chromosomes 8–10. Specifically, the original predictions focused on the central 50kb for each 100kb input sequence. We shuffled the sequences in the first and last 25kb regions to study the effect of flanking regions on the predictions. The results are provided in Table A6 and  A7. Across all settings, we observe that the prediction accuracy slightly decreased with shorter contexts.

| Models | CMP | | | | |
|---|---|---|---|---|---|
| | HAP1 | Hela | HepG2 | K562 | Avg |
| Expert Model | **0.196** | **0.223** | **0.198** | **0.175** | **0.198** |
| CNN | 0.018 | 0.025 | 0.021 | 0.003 | 0.017 |
| HyenaDNA | -0.062 | 0.103 | 0.094 | 0.065 | 0.049 |
| Caduceus-Ph | 0.063 | 0.168 | 0.178 | 0.004 | 0.103 |
| Caduceus-PS | 0.063 | 0.170 | 0.178 | 0.051 | 0.115 |

**Table A11:** Stratum-adjusted correlation coefficient (SCC) for contact map prediction (CMP) task on additional dataet. HAP1, Hela, HepG2, K562 represent different human cell types. We highlight the highest scores in bold. Avg means the average score across different cell types.

## G  RESULTS ON ADDITIONAL ENHANCER-TARGET GENE PREDICTION DATASET

The results for two additional enhancer-target gene prediction datasets are presented in Table A8 and Table A9. These findings highlight that Caduceus-Ph and the expert model gABC achieve the highest performance on their respective datasets.

## H  RESULTS ON ADDITIONAL CONTACT MAP PREDICTION PREDICTION DATASET

The results on the additional contact map prediction task are provided in Table A11. It shows HyenaDNA achieves the highest SCC score among all models.

