# OpenReview forum: "DNALONGBENCH: A Benchmark Suite For Long-Range DNA Prediction Tasks"
_ICLR.cc/2025/Conference — Submitted to ICLR 2025_

### Official Review · Reviewer_kFm8 · 2024-10-29

**Soundness:** 3
**Presentation:** 3
**Contribution:** 2
**Rating:** 6
**Confidence:** 4

**Summary:**

The paper proposes DNALongBench, a new benchmark for genomic tasks that consider long-range dependencies of up to 1 million base pairs. The benchmark provides five tasks, enhancer-target gene interaction, expression quantitative trait loci, 3D genome organization, regulatory sequence activity, and transcription initiation signal, with different requirements. The authors evaluate five models per task, including a task-specific expert model, a CNN baseline and three DNA foundation models. The authors claim that DNALongBench is the most comprehensive benchmark tailored to DNA long-range tasks to date.

**Strengths:**

- The paper studies a significant problem in computational biology. DNA regulation and Chromatin structure appears to be one of the challenges in the field and improvements could lead to novel insights into gene regulation and cellular regulatory networks.

- The paper is well-written and well structured.

- The effort of collecting data from different sources, curating the data, and providing it for download is a strong part of the work.

- Each individual task, the biological importance, the data, and evaluation is well-described. The authors seem to be well informed about the literature and the underlying problem.

**Weaknesses:**

Major:

1. It is unclear to me how many of the tasks really require long context to achieve strong results.

2. A clear limitation of the work, which is also stated by the authors in the conclusion, is that they do not evaluate transformer based models on the different tasks.

3. The codebase is not well documented, there is only a readme and one line referring to a notebook with code for the dataloaders.


Please see more explanations below.

Regarding 1:

Some overview tables/plots in the appendix that clearly show the long-range dependencies of the tasks would make sense I think.


Regarding 2:

While it makes sense that models for genomic data require long contexts, the argument of *NOT* training attention-based models appears weak. The authors use a CNN baseline which obviously does not have a receptive field spanning millions of tokens / base pairs. However, even this “lightweight” baseline seems to outperform the foundation models in some of the tasks (see e.g. Table 4 RSAP Human and Avg, Table 5 WB, SNSES, MS). I would rather say that models with limited context size would be very useful to argue in favor of the development of a long-range benchmark if they really perform worse than non-attention based FMs. Otherwise, the benchmark might still add value, but probably the line of argumentation should be different.


Regarding 3:

I think a benchmark should seamlessly integrate with developer code for a new method to be useful in practice. The authors state that they envision that DNALongBench could be a valuable resource for future evaluations, however, this requires clear documentation, a user friendly interface, and in the best case a leaderboard. A benchmark should integrate into my own codebase with only a few lines of code but it seems like I would have to integrate my code into DNALongBench instead. This clearly limits the usability of DNALongBench and I strongly recommend that the authors work on their code base.

For example, an API as follows would be very helpful to increase the usability

```
import myModel
import Benchmark

benchmark = Benchmark(task=’someTask’)
model = myModel()

def prediction_wrapper(task):
    prediction = model.inference(task.x)
    return prediction

results = benchmark(prediction_wrapper)

print(results)
```
The provided dataloaders might be useful, however, I'm missing a clear description and interface for the evaluation.
Overall, the lack of documentation is an important weakness here. While the API could still be improved until the CRC deadline, I hope the authors can already show some improvements during time of rebuttal.

For a good example benchmark see e.g. [1] (featured paper at NeurIPS 2022 DBT)

Minor:

Line 363 typo: tetails -> details

Line 303: Maybe I missed it but SNPs are not introduced.

Line 479: Additional the

[1] https://openreview.net/forum?id=_HLcjaVlqJ

**Questions:**

See also weaknesses.

- How much of the tasks really require long-range interaction predictions?
- How do attention-based models perform on the tasks?

---

> ### Author Response · Authors · 2024-11-21
> **Response to reviewer's concerns**
>
> We appreciate the reviewer's recognition of the significance of our paper in addressing key challenges in computational biology, particularly in the area of DNA sequence foundation model evaluation. Below, we provide a point-to-point response detailing how we have improved the manuscript and address concerns raised by the reviewers.
>
> **Weakness1 & Q1: It is unclear to me how many of the tasks really require long context to achieve strong results.**
>
> **R1:** We thank the reviewer for raising this important question. We evaluate five tasks under different input context lengths. **We found that all of them indeed requires long contexts.** We will explain our findings on each task.
>
> For the **CMP task**: using Caduceus-Ph as an example, we evaluated the model's performance with input sizes of 409,600, 307,200, and 204,800 base pairs (bps), corresponding to 200, 150, 100 bins, respectively. The evaluation metric was the stratum-adjusted correlation coefficient (SCC). Our results show a clear trend: as context length decreased, prediction performance declines.
>
> |            CMP            | HFF   | H1hESC | GM12878 | IMR90 | HCT116 | Average |
> |------------------------|-------|--------|---------|-------|--------|---------|
> | Caduceus-Ph-409600bps  | 0.153 | 0.130  | 0.101   | 0.138 | 0.145  | 0.133   |
> | Caduceus-Ph-307200 bps | 0.066 | 0.076  | 0.051   | 0.054 | 0.149  | 0.079   |
> | Caduceus-Ph-204800 bps | 0.082 | 0.090  | 0.047   | 0.053 | 0.146  | 0.083   |
>
> For the **RISP task**: using HyenaDNA as an example, we evaluated this task with two different lengths: 196k and 131k bps.
>
> |        RISP             | Human | Mouse |
> |---------------------|-------|-------|
> | Context Length=196k | 0.298 | 0.396 |
> | Context Length=131k | 0.268 | 0.393 |
>
> For the **ETGP task**: instead of simply reducing the input size, we preserved the enhancer sequences and those near gene promoter regions, as these regions are critical for interactions. To assess the impact of context length, we randomly shuffled the central portion of the input sequence (50% of the original length). Using Caduceus-Ph as an example, we observe a notable decline in performance when this central region was shuffled, highlighting the importance of the DNA sequence between enhancers and gene promoters. This is consistent with previous observations [1].
>
> |                                | ETGP  |
> |--------------------------------|-------|
> | Caduceus-Ph                    | 0.826 |
> | Caduceus-Ph w/ shuffled context | 0.418 |
>
> [1] Whalen, S., Truty, R. M., & Pollard, K. S. (2016). Enhancer–promoter interactions are encoded by complex genomic signatures on looping chromatin. Nature Genetics, 48(5), 488-496.
>
> For the **eQTLP task**: similar to ETGP, shuffling the central sequence of Caduceus-Ph led to a decline in performance, indicating that long sequence contexts benefit prediction accuracy.
>
> | eQTLP                          | CCF   | WB    | Thyroid | SNSES | SSELL | MS    | NT    | AT    | AS    | Average |
> |--------------------------------|-------|-------|---------|-------|-------|-------|-------|-------|-------|---------|
> | Caduceus-Ph                    | 0.597 | 0.594 | 0.527   | 0.586 | 0.574 | 0.538 | 0.588 | 0.547 | 0.541 | 0.565   |
> | Caduceus-Ph w/ shuffled context | 0.519 | 0.515 | 0.507   | 0.533 | 0.539 | 0.521 | 0.542 | 0.507 | 0.504 | 0.521   |
>
> For the **TISP task**: using the expert model Puffin-D, we assessed performance with and without short contexts on holdout chromosomes 8–10. For the original input, predictions focused on the central 50kb within each 100kb sequence. To study the impact of flanking regions, we shuffled sequences in the first and last 25kb. Across all settings, we observed a slight performance decline with shorter contexts.
>
> **On the validation chromosome 10:**
>
> |             TISP                  | FANTOM CAGE | ENCODE CAGE | ENCODE RAMPAGE: | GRO-cap | PRO-cap | Average |
> |-------------------------------|-------------|-------------|-----------------|---------|---------|---------|
> | Puffin-D                      | 0.593       | 0.331       | 0.294           | 0.249   | 0.394   | 0.372   |
> | Puffin-D with shorter context | 0.530       | 0.283       | 0.450           | 0.212   | 0.331   | 0.361   |
>
> **On the test chromosomes 8-9:**
>
> |              TISP                 | FANTOM CAGE | ENCODE CAGE | ENCODE RAMPAGE: | GRO-cap | PRO-cap | Average |
> |-------------------------------|-------------|-------------|-----------------|---------|---------|---------|
> | Puffin-D                      | 0.808       | 0.710       | 0.749           | 0.624   | 0.774   | 0.733   |
> | Puffin-D with shorter context | 0.781       | 0.699       | 0.715           | 0.604   | 0.702   | 0.700   |

---

> ### Author Response · Authors · 2024-11-21
> **Rebuttal by authors**
>
> **Continuing answering Weakness 1 & Q1**
>
> In summary, our findings indicate that shorter sequence contexts result in diminished performance across all five tasks. The extent of this performance decline appears to be task-specific. For instance, tasks like enhancer target-gene prediction and contact map prediction show a significant decrease in performance. These outcomes align with the current understanding, emphasizing how regulatory elements rely on long-range interactions for various biological processes. We hope our benchmark will serve as a valuable resource for researchers, enabling the development of model architectures that effectively capture long-range sequence dependencies.
>
> **Weakness 2 & Q2: A clear limitation of the work, which is also stated by the authors in the conclusion, is that they do not evaluate transformer based models on the different tasks.**
>
> **R2:** We thank the reviewer for highlighting this limitation. All datasets included in our benchmarks require input lengths of at least 100kb, with the contact map prediction task necessitating input lengths of up to 1M. Fine-tuning traditional transformer-based models, such as Nucleotide Transformer or DNABERT-2, on these tasks is nearly impossible due to their inherent input length constraints and computational demands. While strategies to extend the context window of transformer-based models and fine-tune them for long-range genomic tasks represent an exciting avenue for future research, we believe that this is beyond the scope of our benchmark study. Our primary goal in this work is to provide biologically meaningful long-range benchmark tasks for evaluating long-range modeling capabilities. We hope that our benchmark will inspire the community to address this important challenge and develop transformer-based approaches capable of handling these tasks in DNA sequences. We hope this clarifies the reviewer's concern.
>
> **Weakness 3: The codebase is not well documented, there is only a readme and one line referring to a notebook with code for the dataloaders.**
>
> **R3**: We appreciate the reviewer’s comment regarding our codebase. We have updated the repository (**https://anonymous.4open.science/r/DNALongBench-FB1D**) to include more guidance for using our benchmark. Additionally, **we have introduced a pyporject.toml file to streamline the setup and provide an API for easier integration.**
>
> **Weakness 4: While it makes sense that models for genomic data require long contexts, the argument of NOT training attention-based models appears weak. The authors use a CNN baseline which obviously does not have a receptive field spanning millions of tokens / base pairs. However, even this “lightweight” baseline seems to outperform the foundation models in some of the tasks (see e.g. Table 4 RSAP Human and Avg, Table 5 WB, SNSES, MS). I would rather say that models with limited context size would be very useful to argue in favor of the development of a long-range benchmark if they really perform worse than non-attention based FMs. Otherwise, the benchmark might still add value, but probably the line of argumentation should be different.**
>
> **R4:** We thank the reviewer for pointing this out. First, due to current computational resource limitations, achieving an attention-based model capable of processing millions of tokens is infeasible. Second, as outlined in our responses to Weakness 1 & Q1, reducing context size consistently resulted in performance degradation across all five tasks. This highlights the critical role of long-range context dependencies in addressing these biologically meaningful challenges. Thus, we believe our benchmark serves as a meaningful evaluation framework to advance long-range genomic modeling.
>
> **Weakness 5: To make DNALongBench could be a valuable resource for future evaluations, a clear documentation, a user friendly interface, and in the best case a leaderboard.**
>
> **R5:** We appreciate the reviewer’s suggestions. We have incorporated your feedback by updating our GitHub repository documentation, improving the user interface, and creating a leaderboard website. We simplified the DNALongBench installation process (**https://anonymous.4open.science/r/DNALongBench-FB1D**) to make it more accessible. **Users can now import the DNALongBench package and select tasks to evaluate via an API interface**. Additionally, **we launched a leaderboard website (https://dnalongbench.github.io/DNALongBench)** to demonstrate the performance of different models.
>
> **Minor issues: Line 363 typo: tetails -> details, Line 303: Maybe I missed it but SNPs are not introduced, Line 479: Additional the**
>
> **R6:** We have corrected these minor issues. Please refer to the revised manuscript highlighted in red in our uploaded version: (1) line 363 we fixed the typo. (2) line 215, we replaced SNPs with single-nucleotide polymorphisms (SNPs). (3) We didn’t find “Additional the” in line 479.

---

> > ### Comment · Reviewer_kFm8 · 2024-11-21
> > **Response to author rebuttal**
> >
> > I acknowledge the efforts of the authors to improve the code base. Further, I'm happy with the analysis provided by the authors regarding the importance of long-range interactions in the benchmark. Having looked at the revised version of the manuscript, I would still recommend to also include these observations in the appendix. I think these are valuable insights that could be of interest to the community. However, this is my personal opinion and I would leave it to the authors to decide about further updates of the manuscript. The authors addressed my concerns and questions and I have decided to revise my score to reflect the progress made.

---

> > > ### Author Response · Authors · 2024-11-21
> > > **Response to reviewer kFm8**
> > >
> > > We sincerely appreciate the reviewer’s response and are happy that we have addressed the reviewer’s concerns. According to the reviewer’s further suggestions, we have included all the results into our appendix. Please refer to the Appendix F and Introduction line 88 highlighted in red.

---

### Official Review · Reviewer_FfkD · 2024-10-31

**Soundness:** 3
**Presentation:** 3
**Contribution:** 3
**Rating:** 6
**Confidence:** 5

**Summary:**

This paper presents DNALongBench, which consists of 5 datasets which attempt to test a predictive model’s ability to make predictions on DNA sequences which are somewhat related to long-range interactions in the genome. Each of these datasets consists of a set of inputs and outputs, along with a well-defined performance metric. The input is a long DNA sequence and the output is a label which can be a scalar, or one or more functional-readout profiles which are parallel to the input sequence.

These 5 datasets and tasks are: 1) given a sequence with an enhancer and a promoter in it, predict whether or not the enhancer is linked to the promoter functionally; 2) given a long sequence, predict the (symmetric) 2D contact map; 3) given a DNA sequence, predict the functional profiles of several regulatory-genomic assays in both human and mouse; 4) given the reference and mutated sequences of putative eQTLs, predict whether or not the eQTL is causal/functional; 5) given a DNA sequence, predict base-pair-level profiles of transcription initiation.

For each task, the authors computed the performance of the dataset on HyenaDNA and Caduceus, which have been fine-tuned for each task separately. They also trained a simple CNN as a baseline, and compared to an “expert model”, which is the model which was trained specifically for that task. The authors report the resulting performance metrics, and describe some general trends, such as the observation that the expert model generally performed the best, and that the long-range language models (e.g. HyenaDNA) typically underperformed.

**Strengths:**

### Good test of actual functional abilities of long-range DNA language models

In the recent literature, a lot of effort has been made to port over LLMs from NLP to DNA sequences (and other biological tasks). In these works, the evaluation of DNA LLMs has been very sparse, and it was unclear as to whether or not these models could handle real biological tasks of significance (instead of very simplistic predictive tasks like species prediction, which is not biologically meaningful or challenging today as a computational task).

This paper offers a much-needed evaluation of whether or not these DNA LLMs can actually handle biologically meaningful and challenging tasks which rely on long-range interactions (which was the initial promise of these models). Through various experiments, this paper shows that, in fact, these DNA LLMs are not particularly well-poised to predict meaningful biological tasks, even after fine-tuning.

### Good selection of benchmarks with meaningful biological significance

The selection of the 5 benchmark tasks is well-thought-out, and they span many areas of interest, from regulatory genomics, to chromatin organization, to variant-effect prediction (for eQTLs). Together, these tasks cover a decent space in testing the ability of models to capture trans effects in the genome.

### Clear writing and well-organized paper

The paper is well-written, with clear organization and motivation. Reading through, things were simple to understand, and structured in a very good way.

**Weaknesses:**

### Other than large DNA language models, not many other models are tested

The DNA LLMs that were tested (i.e. HyenaDNA and Caduceus) are good representations of this sort of model, but there aren’t that many other models that were tested in this benchmark. Other than these DNA LLMs, this work trains a very simplistic CNN, and applies a single expert model (generally, the model which was published along with the data). Together, the expert model is an example of one of the best models possible, and the CNN is an example of one of the easiest deep-learning-based baselines. The analyses in this paper show how DNA LLMs perform relative to these two endpoints, but there are not any other models that are tested.

In addition to these DNA LLMs, many researchers will use other models and architectures which can also detect long-range interactions (such as those described in section 2.2). Especially due to the computational cost of DNA LLMs, many researchers will necessarily be relying on other architectures. It would be much more informative to show the performance of these other architectures (e.g. Enformer, Borzoi, even DanQ, etc., both fine-tuned and not fine-tuned on the specific task), to see how these models also perform. In its current form, this paper is largely just a benchmark for DNA LLMs, even though several other models exist, whose performance also falls somewhere in the spectrum between the basic CNN and the expert model. It would be far more meaningful of a benchmark if those models are shown, as well.

### Limited results on the various advantages and disadvantages of certain models

There are only limited results in the paper exploring and discussing the relative advantages and disadvantages of certain models. For a benchmarking paper, it would be useful to have more discussion on the various areas where certain models do better than others, and an exploration of why (including different ablation studies). Some interesting discussion points (a very non-comprehensive set of suggestions) may be:

- How useful is reverse-complement augmentation or reverse-complement-aware architectures for these tasks?
- Since the expert model really does much better, how much of its improved performance is due to its focus on a specific task (whereas the DNA LLMs are not trained on a specific task)? How much does fine-tuning the DNA LLMs allow them to perform much better on these specific tasks?
- How much performance is gained from the sheer size/expressivity of the DNA LLMs? Would a different architecture of similar size/expressivity be comparable to the DNA LLMs (e.g. testing an Enformer-like architecture of high capacity, for each of these tasks).
- Section 5.2 suggests a difference between long-range and short-range prediction performance in these models. Is the failure of DNA LLMs to predict specific peaks due to the excessively long input sequences (i.e. so the model has a hard time focusing on specific parts of a sequence)?

### Suggestions for improvements in writing

There are several places where the paper has some typos or other grammatical errors. Here is a non-comprehensive list:

- An en-dash (not a hyphen, and not an em-dash) should be between words/objects which specify a pair: enhancer–target gene, enhancer–promoter, variant–promoter
- “megabase pairs” should be “mega basepairs” or “mega base-pairs” (L220)
- “complied” should be “compiled” (L278)
- “tetails” should be “details” (L363)

Additionally, there are areas which could use more clarity

- In the tables comparing performance between the models, because the expert model is effectively always by far the best one, it would be useful to underline the second-best model, as well (or use another way to emphasize the second-best model in addition to the best)
- In Table 2, the shape of the output in contact-map prediction is a bit confusing, as contact maps are typically thought of as 2D objects rather than a single 99681-vector
- The order of the datasets/tasks is not the same in the Table 2, the main text, and the supplement in different sections; there doesn’t seem to be a reason why the ordering of these 5 tasks can’t be presented in the same ordering each time
- In the data repositories at `https://dataverse.harvard.edu/`, it is not clear what each file represents and how the data is organized *vis a vis* the descriptions in the appendix; a README should be added to each repository

**Questions:**

1. What is the stratified sampling approach used in some of these tasks (i.e. enhancer–target-gene prediction, eQTL prediction); that is, what is the stratification based off of?
2. In any of these datasets/tasks, is it possible that the same genomic coordinate appears in multiple dataset splits (e.g. train and test)? This could be possible if the examples arising from the same gene end up in different splits, for example
3. When masking out other enhancers in the enhancer–target-gene task, what were these enhancers masked with?
4. What was the reasoning for removing genes with too few positive/negative pairs (i.e. enhancer–target-gene prediction, eQTL prediction)?

---

> ### Author Response · Authors · 2024-11-21
> **Rebuttal by authors**
>
> We appreciate the reviewer’s thoughtful and constructive questions. We are particularly encouraged that you found our work is a much-needed evaluation of long-range DNA LLMs’ ability to biologically meaningful and challenging tasks. Our answers to specific points are shown below:
>
> **Weakness1: In addition to these DNA LLMs, many researchers will use other models and architectures which can also detect long-range interactions (such as those described in section 2.2). Especially due to the computational cost of DNA LLMs, many researchers will necessarily be relying on other architectures. It would be much more informative to show the performance of these other architectures (e.g. Enformer, Borzoi, even DanQ, etc., both fine-tuned and not fine-tuned on the specific task), to see how these models also perform. In its current form, this paper is largely just a benchmark for DNA LLMs, even though several other models exist, whose performance also falls somewhere in the spectrum between the basic CNN and the expert model. It would be far more meaningful of a benchmark if those models are shown, as well.**
>
> **R:** We thank the reviewer for raising this important question. One of the primary goals of this work is to provide a set of biologically meaningful long-range genomic tasks and use them to evaluate the performance of current DNA LLMs. DNA LLMs are promising but current evaluation benchmarks mostly focus on short-range genomic tasks or limited biological data types. We first compiled five long-range genomic data for benchmarking. To make an effective comparison, for each task, we selected one of the best models specifically designed to each task. These expert models actually represent a wide-spread type of model architecture. For instance, we evaluated  the Enformer model, a CNN and transformer hybrid model, on the regulatory sequence activity prediction (RSAP) and eQTL prediction (eQTLP) task. Puffin-D, a CNN based model with two upward-downward passes and residual connections similar to U-net, was used to evaluate the transcription initiation signal prediction (TISP) task. Regarding the enhancer target-gene prediction (ETGT) task, we chose the activity-by-contact (ABC) model. This model is simple but has been widely shown with good performance against more complicated deep learning models. For instance, the authors in the gABC model [1] have found the gABC model shows better performance than Enformer. Note that we also included gABC as the expert model in two additional CRISPRi screening-based enhancer-target gene datasets, of which the performance is shown in the following tables.  As suggested by the reviewer, it would be quite interesting to compare the performance of the above-mentioned best models on each task, but this deviates from our primary goal to evaluate the DNA LLMs. In response to the reviewer, we have now conducted a series of ablation experiments to explore the advances and limitations of DNA LLMs (also see our response to reviewer kFm8). For instance, we found that context lengths indeed show great impact on the model performance, suggesting DNA LLMs can learn some long-range dependency. However, the relatively poor performance of DNA LLMs as compared with these best models indicate that there is still a lot of room for the development of DNA LLMs. Model size is important but is not deterministic (see the response to weakness 2.3). We hope this response can answer the reviewer's questions.
>
> [2] Schraivogel et al.
>
> | Model               | ETGP  |
> |---------------------|-------|
> | Expert Model (ABC)        | 0.896 |
> | Expert Model (gABC) | 0.890 |
> | CNN                 | 0.711 |
> | HyenaDNA            | 0.752 |
> | Caduceus-Ph         | 0.906 |
> | Caduceus-ps         | 0.895 |
>
> [3] Gasperini et al.
>
> | Model              | ETGP  |
> |--------------------|-------|
> | Expert Model (ABC) | 0.824 |
> | Expert Model (gAB) | 0.831 |
> | CNN                | 0.730 |
> | HyenaDNA           | 0.781 |
> | Caduceus-Ph        | 0.788 |
> | Caduceus-ps        | 0.793 |
>
> [1] Hecker, D., Behjati Ardakani, F., Karollus, A., Gagneur, J., & Schulz, M. H. (2023). The adapted Activity-By-Contact model for enhancer–gene assignment and its application to single-cell data. Bioinformatics, 39(2).
>
> [2] Targeted Perturb-seq enables genome-scale genetic screens in single cells. Schraivogel et al. Nature Methods. 2020.
>
> [3] A genome-wide framework for mapping gene regulation via cellular genetic screens. Gasperini et al. Cell. 2019.

---

> ### Author Response · Authors · 2024-11-21
> **Rebuttal by authors**
>
> **Weakness 2.1: How useful is reverse-complement augmentation or reverse-complement-aware architectures for these tasks?**
>
> **R:** We appreciate the reviewer’s question regarding the usefulness of reverse-complement augmentation and reverse-complement-aware architectures for these tasks. To address this, we first evaluated the effect of reverse-complement (RC) augmentation using Caduceus-Ph on the enhancer target-gene prediction (ETGP) task. We also implemented shift augmentation by randomly shifting the input by 1-3 base pairs (bps) to either 5’ and 3’ end of the DNA sequence, as this is another common practice in the field to augment data. According to our results as shown below, we observed that applying reverse-complement or shift augmentations led to comparable or slightly worse performances for Caduceus-Ph compared to training without these augmentations. We think the reason might be because applying augmentations may introduce additional complexity in the patterns of positive samples for models to learn. Furthermore, promoter-enhancer orientation is important for this task. The input sequence is oriented with the gene positioned at the 5’ end and the enhancer located at the 3’ end. And this task requires orientation-specific information, so reverse-complement augmentation might not be that useful in this task.
>
> |      | Expert Model | CNN   | Caduceus-Ph | Caduceus-Ph-with shift data augmentation | Caduceus-Ph-with  complementary data augmentation  |
> |------|--------------|-------|-------------|------------------------------------------|----------------------------------------------------|
> | ETGP | 0.926        | 0.797 | 0.826       | 0.821                                    | 0.820                                              |
>
> **Weakness2.2: Since the expert model really does much better, how much of its improved performance is due to its focus on a specific task (whereas the DNA LLMs are not trained on a specific task)? How much does fine-tuning the DNA LLMs allow them to perform much better on these specific tasks?**
>
> **R2.2:** We have already finetuned the DNA LLMs on each of the classification or regression tasks. Therefore, the performance of each model is achieved by training or tuning the model on a specific task. We think that the performance gap between expert model and DNA LLMs is likely due to the fact that current DNA LLMs still lack the ability to fully capture the complex dependencies across long-range contexts. We hope our benchmark data could be a good resource for researchers to test various model architectures to more effectively learn long-range sequence dependencies.
>
> **Weakness 2.3: How much performance is gained from the sheer size/expressivity of the DNA LLMs? Would a different architecture of similar size/expressivity be comparable to the DNA LLMs (e.g. testing an Enformer-like architecture of high capacity, for each of these tasks).**
>
> **R2.3:** In our setting, we utilize Enformer as the expert model for the eQTLP, ETGP, and RSAP tasks. The model sizes for Enformer and DNA LLMs are reported below. Notably, Enformer has a significantly larger model size and higher capacity compared to HyenaDNA and Caduceus. However, previous literature suggests that larger model size alone does not always lead to a better performance. For instance, in the Caduceus paper [1], the authors found that DNABERT-2, despite its larger size (117M parameters), underperforms compared to Caduceus in several long-range genomic benchmark tasks. This highlights the critical importance of architectural design tailored to the specific task, which has a substantial impact on final performance outcomes.
>
> |            | Enformer | HyenaDNA | Caduceus |
> |------------|----------|----------|----------|
> | Model Size | 252M     | 1.6M     | 1.9M     |
>
> [1] Schiff, Y., Kao, C. H., Gokaslan, A., Dao, T., Gu, A., & Kuleshov, V. (2024). Caduceus: Bi-directional equivariant long-range dna sequence modeling. arXiv preprint arXiv:2403.03234.

---

> > ### Author Response · Authors · 2024-11-21
> > **Rebuttal by authors**
> >
> > **Weakness 2.4: Section 5.2 suggests a difference between long-range and short-range prediction performance in these models. Is the failure of DNA LLMs to predict specific peaks due to the excessively long input sequences (i.e. so the model has a hard time focusing on specific parts of a sequence)?**
> >
> > **R2.4:** We thank the reviewer for this interesting question. To answer this question, we evaluate DNA LLMs under different input sequence lengths. We chose the contact map prediction task as it is known that there exist chromatin interaction patterns at different length scales. We evaluate the contact map prediction task under three input sizes, i.e. 409600, 307200 and 204800 bps. We used stratum-adjusted correlation coefficient (SCC) to evaluate the prediction. We found that there is a performance decline when we reduce the context length, indicating that though there is a performance gap between the expert model Akita and DNA LLMs, utilizing long-range sequences still helps to obtain a better performance for DNA LLMs.
> >
> > **Contact map prediction with different context length:**
> >
> > **Context length 409600 bps**
> >
> > |             | HFF   | H1hESC | GM12878 | IMR90 | HCT116 | Average |
> > |-------------|-------|--------|---------|-------|--------|---------|
> > | CNN         | 0.025 | 0.024  | 0.010   | 0.013 | 0.001  | 0.015   |
> > | Caduceus-Ph | 0.153 | 0.130  | 0.101   | 0.138 | 0.145  | 0.133   |
> >
> > **Context length 307200 bps**
> >
> > |             | HFF   | H1hESC | GM12878 | IMR90 | HCT116 | Average |
> > |-------------|-------|--------|---------|-------|--------|---------|
> > | CNN         | 0.029 | 0.032  | 0.012   | 0.022 | 0.007  | 0.020   |
> > | Caduceus-Ph | 0.066 | 0.076  | 0.051   | 0.054 | 0.149  | 0.079   |
> >
> > **Context length 204800 bps**
> >
> > |             | HFF    | H1hESC | GM12878 | IMR90  | HCT116 | Average |
> > |-------------|--------|--------|---------|--------|--------|---------|
> > | CNN         | -0.003 | 0.049  | 0.012   | -0.006 | 0.003  | 0.011   |
> > | Caduceus-Ph | 0.082  | 0.090  | 0.047   | 0.053  | 0.146  | 0.083   |
> >
> > **Suggestions for writing: There are several places where the paper has some typos or other grammatical errors. Additionally, there are areas which could use more clarity.**
> >
> > **R:** Thanks for pointing out these typos and errors. We have revised the manuscript accordingly and uploaded the new version of the manuscript. Please refer to the revised manuscript highlighted in red: (1) L220, “mega basepairs”, (2) L 278, “compiled”, (3) L 363,”details”.
> >
> > **Q1: What is the stratified sampling approach used in some of these tasks (i.e. enhancer–target-gene prediction, eQTL prediction); that is, what is the stratification based off of?**
> >
> > **R1:** We apologize for any confusion. In both enhancer target-gene and eQTL predictions, our dataset exhibits a significant class imbalance. This occurs because true enhancer-gene pairs constitute only a small fraction of the pairs tested. To address this, we employed stratified sampling rather than random sampling. This approach prevents the scenario where random sampling might allocate all negative samples to either the training or validation set, which would result in biased model training and evaluation. Specifically, we first screened out genes with fewer than two positive and two negative enhancer-gene pairs. While creating the training, validation, and test sets, stratified sampling ensures that each of the training and validation sets contains at least one positive and one negative sample. The remaining samples are then randomly divided among the training, validation, and test sets. We implemented the same strategy for eQTL prediction for similar reasons.
> >
> > **Q2: In any of these datasets/tasks, is it possible that the same genomic coordinate appears in multiple dataset splits (e.g. train and test)? This could be possible if the examples arising from the same gene end up in different splits, for example**
> >
> > **R2:** We thank the reviewer for the question. We agree with the reviewer that it is important to make sure there is no information leakage between training and test set. For the enhancer target-gene task, each gene has its own enhancer candidates within 450 kb from the transcription start sites. The input of the model is the sequence between the promoter of the gene and the enhancer candidate. The enhancer-gene interactions are mostly gene-specific so holding out some genes in the training set might be very informative.  To avoid the scenarios where an enhancer candidate in the training set may also exist in the input sequence of the test enhancer, we masked all enhancer candidates by letter N in the input sequence.
> >
> > **Q3: When masking out other enhancers in the enhancer–target-gene task, what were these enhancers masked with?**
> >
> > **R3:** These enhancer input sequences are masked with the other enhancer candidates of its potential target genes. The reason is to eliminate the confounding effects that might arise from the presence of other enhancer candidates.

---

> ### Author Response · Authors · 2024-11-21
> **Rebuttal by authors**
>
> **Q4: What was the reasoning for removing genes with too few positive/negative pairs (i.e. enhancer–target-gene prediction, eQTL prediction)?**
>
> **R4:** The rationale is to ensure that the model has sufficient data to learn the sequence dependency between an enhancer and its target genes. Given the limitations of CRISPR screening, not all potential enhancer candidates for a gene are experimentally tested. As a result, we have decided to concentrate solely on genes that have an adequate number of enhancer candidates and verified positive pairs.
>
> ==
>
> We believe that our responses have effectively addressed the reviewer's concerns. If there are any further questions or suggestions, we would be more than happy to discuss them. We look forward to any additional feedback the reviewer may provide.

---

> ### Author Response · Authors · 2024-11-25
>
> Dear reviewer FfkD,
>
> We sincerely appreciate the time and effort you've dedicated to reviewing our paper. In our response, we have carefully and diligently addressed your thoughtful concerns and insightful questions. As the discussion deadline is approaching, please do not hesitate to let us know if you have any additional questions or need further clarification. Your insights and feedbacks are very important for us to improve our work.
>
> Best,
> Authors of paper 5418

---

### Official Review · Reviewer_4eai · 2024-11-01

**Soundness:** 3
**Presentation:** 3
**Contribution:** 2
**Rating:** 5
**Confidence:** 4

**Summary:**

This paper proposes a set of five benchmarking tasks for genomic language models, with a focus on tasks that involve modeling long-range DNA dependencies, including enhance-target gene interaction and 3D genome organization. These tasks represent a diverse set of biologically important characteristics. The authors benchmark five baseline methods on each task - task-specific expert models, CNN-based models, and three recent DNA language models, and find that the DNA language models they benchmark do not outperform the task-specific models.

**Strengths:**

- The tasks included in the benchmark are biologically significant and diverse, especially the focus on tasks with different dimensionalities (1D or 2D) and granularities (binned or base pair level). In addition, the paper does a good job explaining the biological significance of each task. Both the increased focus on long-range tasks and base pair resolution tasks represent novel aspects of this benchmark compared to other previously published benchmarks.
- The results are presented in a clear and concise manner
- The performance of the three evaluated DNA language models — HyenaDNA, Caduceus-Ph, and Caduceus-PS — on these tasks reveal important limitations of current DNA language models at modeling long range dependencies, even when the models are able to incorporate long context lengths.

**Weaknesses:**

- In my opinion, the main weakness of this paper is novelty. Although the benchmarking tasks are important and biologically motivated, all five of the tasks and corresponding datasets have already been used to benchmark models in previous publications and in some previous benchmarks, such as BEND (Marin, et al. 2023) and LRB (Kao, et al. 2024). The “Regulatory sequence activity” task directly uses the training, validation, and test set sequences from Enformer. Therefore, the main contribution of this paper is consolidating these tasks into one resource and comparing the performance of DNA language models and more traditional supervised approaches. One potential way to increase the novelty and contribution of this work could be to incorporate more datasets that weren’t used in previous publications to increase the amount of benchmarking data for each task. For example, for the enhancer-target gene task, the dataset is relatively small, and different experimental datasets often suffer from their own technical and experimental biases. Since a number of similar datasets exist and are publicly available, it would improve and increase the impactfulness of the benchmark to aggregate multiple similar datasets.

**Questions:**

- The term “Expert model” is misleading. For example, L369-370 refers to “expert models tailored to each task,” but in some cases, such as using Enformer as an expert model for eQTL prediction, the expert model has not actually been tailored to this task. An alternative could be to refer to these models at “State of the art models.” In addition, in some cases the choice of Expert/SOTA model could be revisited. For example, for the enhancer-target gene task, Enformer (Avsec et al. Nature Methods (2021)) or Borzoi (Linder et al. 2023) should be used as a state of the art model instead of Activity by Contact.  In addition, for the eQTL prediction task, Borzoi was shown to outperform Enformer in Linder et al. 2023.
- Use of the term DNA foundation model and which models are classified in this category is inconsistent throughout the paper. For example, on L146, Avsec et al. 2021a is cited as a DNA foundation model (although it is not actually pre-trained on only DNA seqeunces), but later in the paper it’s used as an “expert model” baseline in contrast to the DNA foundation models.

Minor suggestions:
- L51-53 - Karollus et al. Genome Biology (2023) could be cited with reference to this point
- L154-155 is unclear and should be reworded: “It has shown promising performance in long-range species classification tasks despite the problem itself is not well defined in real applications.”

---

> ### Author Response · Authors · 2024-11-22
> **Rebuttal by authors**
>
> We appreciate the reviewer’s valuable feedback and insightful questions. These questions help improve the overall quality of our manuscript. In response to the reviewer's concerns, we have incorporated additional datasets and conducted further experiments. Our detailed answers to the specific points are provided below:
>
> **Weakness: In my opinion, the main weakness of this paper is novelty. Although the benchmarking tasks are important and biologically motivated, all five of the tasks and corresponding datasets have already been used to benchmark models in previous publications and in some previous benchmarks, such as BEND (Marin, et al. 2023) and LRB (Kao, et al. 2024). The “Regulatory sequence activity” task directly uses the training, validation, and test set sequences from Enformer. Therefore, the main contribution of this paper is consolidating these tasks into one resource and comparing the performance of DNA language models and more traditional supervised approaches. One potential way to increase the novelty and contribution of this work could be to incorporate more datasets that weren’t used in previous publications to increase the amount of benchmarking data for each task. For example, for the enhancer-target gene task, the dataset is relatively small, and different experimental datasets often suffer from their own technical and experimental biases. Since a number of similar datasets exist and are publicly available, it would improve and increase the impactfulness of the benchmark to aggregate multiple similar datasets.**
>
> **R:** We thank the reviewer for the valuable comments. **First, we want to clarify that three of the five benchmark datasets (i.e., enhancer-target gene, contact map, and transcription initiation signal) are not included in any existing benchmarks**, such as BEND and LRB. Enhancer-target gene prediction and contact maps represent well-known regulatory mechanisms involving DNA interactions that span thousands or millions of base pairs. These tasks also introduce new and more challenging task types, including 2D contact map prediction and base pair resolution regression, which were not addressed in any existing benchmarks.
>
> We agree with the reviewer that incorporating more relevant datasets would increase the size of benchmarking data. **To address this, we have collected two additional CRISPRi screening-based enhancer-target gene datasets (line 231-232, line 787-790) and evaluated the performances of different models (Table A8 and Table A9).** CRISPRi screening-based methods are among the most direct and accurate methods to determine whether an enhancer regulates specific target genes. To the best of our knowledge, only three such datasets are publicly available, and we have incorporated all of them into our benchmarking task.
>
> The following table shows the performance of different models on the enhancer target-gene tasks. We observe a consistent trend that expert models including ABC and its variants gABC [1] outperform CNN baseline and three DNA language models. The authors in the gABC model have found that gABC shows higher performance than ABC and Enformer. In our evaluations on the benchmark data, we observed comparable performance between the ABC and gABC models. **These results have been included in the manuscript in Tables A8 and A9, and are summarized in Appendix G.**
>
> **Two additional dataset on enhancer-target gene prediction task**
>
> **Table A8 ETGP results on data curated from Schraivogel et al**
>
> | Model               | ETGP  |
> |---------------------|-------|
> | Expert Model        | 0.896 |
> | Expert Model (gABC) | 0.890 |
> | CNN                 | 0.711 |
> | HyenaDNA            | 0.752 |
> | Caduceus-Ph         | **0.906** |
> | Caduceus-ps         | 0.895 |
>
> **Table A9 ETGP results on data curated from Gasperini et al**
>
> | Model              | ETGP  |
> |--------------------|-------|
> | Expert Model (ABC) | 0.824 |
> | Expert Model (gAB) | **0.831** |
> | CNN                | 0.730 |
> | HyenaDNA           | 0.781 |
> | Caduceus-Ph        | 0.788 |
> | Caduceus-ps        | 0.793 |
>
> [1] Hecker, D., Behjati Ardakani, F., Karollus, A., Gagneur, J., & Schulz, M. H. (2023). The adapted Activity-By-Contact model for enhancer–gene assignment and its application to single-cell data. Bioinformatics, 39(2).
>
> [2] Targeted Perturb-seq enables genome-scale genetic screens in single cells. Schraivogel et al. Nature Methods. 2020.
>
> [3] A genome-wide framework for mapping gene regulation via cellular genetic screens. Gasperini et al. Cell. 2019.

---

> ### Author Response · Authors · 2024-11-22
> **Rebuttal by authors**
>
> **Continuing replying to weakness**
>
> **We also curated and processed four additional datasets for the contact map prediction task (Table A10, line 263-265, and line 817-822).** Briefly, we downloaded Hi-C data for four different cell types from the 4DN data portal and processed them following the same procedure described in the Akita paper. Due to the time constraints, we only evaluated the performance of different models in the 5% downsampled data. We observed some overfitting issues, likely due to the small size of training data. We plan to update the table and leaderboard once the training process is complete. **The data source and results have been included in the manuscript in Tables A10 and A11, and Appendix H.**
>
> **Table A10 data source on additional data set for contact map prediction**
>
> | Cell type | Accession ID on 4DN data portal |
> |-----------|---------------------------------|
> | HAP1      | 4DNFIWGGYEW2                    |
> | Hela      | 4DNFIIT9PKS9                    |
> | HUVEC     | 4DNFI2R1VS8E                    |
> | K562      | 4DNFIIT9PKS9                    |
>
> **Table A11 Performance on additional Dataset on Contact Map prediction**
>
> |              | HAP1    | Hela   | HUVEC    | K562      | Average |
> |--------------|---------|--------|----------|-----------|---------|
> | Expert Model | 0.0100  | 0.0120 | 0.0010   | 0.0010    | 0.0060  |
> | CNN          | 0.023   | 0.0115 | -0.00025 | 0.0008359 | 0.0087  |
> | HyenaDNA     | -0.0307 | 0.342  | 0.0590   | -0.00766  | **0.0906**  |
> | Caduceus-Ph  | 0.0483  | 0.0120 | 0.0163   | 0.0257    | 0.0256  |
> | Caduceus-PS  | 0.0477  | 0.0121 | 0.0154   | 0.0259    | 0.0252  |
>
> **Q1: The term “Expert model” is misleading. For example, L369-370 refers to “expert models tailored to each task,” but in some cases, such as using Enformer as an expert model for eQTL prediction, the expert model has not actually been tailored to this task. An alternative could be to refer to these models as “State of the art models.” In addition, in some cases the choice of Expert/SOTA model could be revisited. For example, for the enhancer-target gene task, Enformer (Avsec et al. Nature Methods (2021)) or Borzoi (Linder et al. 2023) should be used as a state of the art model instead of Activity by Contact. In addition, for the eQTL prediction task, Borzoi was shown to outperform Enformer in Linder et al. 2023.**
>
> **R:**  We thank the reviewer for the valuable suggestions. In response, we have replaced the phrase “expert models tailored to each task” with “expert models that have shown state-of-the-art results” on lines 369–370. However, we opted not to replace all mentions of expert models with SOTA for several reasons, as we believe this term may lead to some confusion.
>
> For example, in the enhancer target-gene task, the Borzoi model was trained to predict gene expression, and the authors of Borzoi demonstrated its ability to predict target genes by evaluating gradient attributions. While this approach appears reasonable, recent reports, including [1] and [2], indicate that such models may underestimate distal enhancers and exhibit asymmetric bias depending on the direction relative to the transcription start site (TSS). Furthermore, a variant of the ABC model, named gABC, has been shown to outperform Enformer [3]. We included gABC as an additional expert model and reevaluated the task using two newly added datasets. Both ABC and gABC demonstrate similar performance levels.
>
> Lastly, we want to highlight that the primary goal of this work is to provide a set of biologically meaningful and challenging long-range genomic tasks and use them to evaluate the performance of current DNA LLMs. While DNA LLMs are promising, existing evaluation benchmarks primarily focus on short-range genomic tasks or limited biological data types. Our evaluations show that the current expert models selected in this study have already outperform DNA LLMs.  Models like Borzoi are still under development and have not yet been published, so we chose to adopt a conservative approach in referring to these models as SOTA.
>
> [1] Karollus, A., Mauermeier, T., & Gagneur, J. (2023). Current sequence-based models capture gene expression determinants in promoters but mostly ignore distal enhancers. Genome biology, 24(1), 56.
>
> [2] Toneyan, S., & Koo, P. K. (2023). Interpreting cis-regulatory interactions from large-scale deep neural networks for genomics. bioRxiv.
>
> [3] Hecker, D., Behjati Ardakani, F., Karollus, A., Gagneur, J., & Schulz, M. H. (2023). The adapted Activity-By-Contact model for enhancer–gene assignment and its application to single-cell data. Bioinformatics, 39(2).

---

> ### Author Response · Authors · 2024-11-22
> **Rebuttal by authors**
>
> **Q2: Use of the term DNA foundation model and which models are classified in this category is inconsistent throughout the paper. For example, on L146, Avsec et al. 2021a is cited as a DNA foundation model (although it is not actually pre-trained on only DNA seqeunces), but later in the paper it’s used as an “expert model” baseline in contrast to the DNA foundation models.**
>
> **R:** We agree with the reviewer. We have fixed the issue by removing Avsec et al 2021a on L146.
>
> **Minor suggestions1: L51-53 - Karollus et al. Genome Biology (2023) could be cited with reference to this point**
>
> **R:** According to the reviewer’s suggestion, we have cited this paper. Please see the reuploaded version, where the citation  has been added on line 53 and is highlighted in red.
>
> **Minor suggestions2: L154-155 is unclear and should be reworded: “It has shown promising performance in long-range species classification tasks despite the problem itself is not well defined in real applications.”**
>
> **R:** We thank the reviewer for pointing this out. We have revised the paper accordingly. The changes are highlighted in red in the reuploaded version on lines 154–155. The sentence has been replaced with: “It has demonstrated promising performance in long-range species classification tasks, even though the practical applications of this problem remain poorly defined.”
>
> ==
>
> We sincerely thank the reviewer for their thoughtful and constructive comments, which have significantly improved the quality and clarity of our paper. We believe that we have thoroughly addressed all the concerns raised and made substantial revisions accordingly. If the reviewer has any additional questions or suggestions, we would be happy to discuss them further. We look forward to any additional feedback the reviewer may have.

---

> > ### Comment · Reviewer_4eai · 2024-11-24
> >
> > Thanks to the authors for their detailed response and updates to the manuscript. The authors have sufficiently addressed my questions regarding choice of expert models and terminology in the manuscript. In addition, the authors have partially addressed my concerns about novelty with the addition of additional datasets for the enhancer-target gene and contact map prediction tasks. While I still believe there is significant overlap between this benchmark and prior benchmarking studies, I have updated my score to reflect the novelty gained by incorporating additional datasets not used in previous evaluations.

---

> ### Author Response · Authors · 2024-11-25
> **Further response to reviewer's concerns**
>
> We sincerely thank the reviewer for their thoughtful feedback and careful review of our responses and manuscript updates. We are encouraged to hear that our revisions have effectively addressed many of your concerns. Regarding your comments on the perceived overlap of benchmark data with BEND and LRB, we would like to take this opportunity to further clarify the distinctions between DNALongBench and these prior benchmarks. As we explain below, **DNALongBench not only incorporates more diverse experimental data types but also provides significantly larger datasets, enabling more comprehensive evaluations compared to BEND and LRB.**
>
> **Comparison with BEND:**
>
> BEND contains seven different classification tasks, five of which involve input DNA sequences limited to 512bp. The input length for gene finding ranges between 1,433 and 14,000 base pairs. **The enhancer annotation task is the only task that can be considered a long-range genomic task with an input length around 100kb, whereas the input length in DNALongBench is up to 1M base pairs. Additionally, this task includes only 285 samples, which is much smaller than DNALongBench. Furthermore, the task goal in BEND differs significantly from DNALongBench.** In BEND, the task is to predict whether each 128 bp bin is an enhancer, without addressing enhancer and its target genes relationships, which are a more challenging core focus of DNALongBench.
>
> **Comparison with LRB:**
>
> LRB consists of seven tasks. Only CAGE and variant effect prediction have some overlap with what we used by deriving from the Enformer datasets. However, both the dataset sizes and task settings in LRB are not comparable with DNALongBench.
>
> For instance, the authors of LRB only selected 50 CAGE tracks from the 638 available in humans. **In contrast, DNALongBench includes a total of 6,991 tracks from human and mouse -- over 100 times more than LRB, as shown in the table below.** This decision was made to ensure a fair and consistent comparison between DNA LMs and a strong supervised baseline (Enformer). While LRB also used Enformer data, their setup included only 50 tracks, with Enformer serving as the task-specific expert model baseline. We believe that providing the full training dataset, as used by Enformer, enables a more comprehensive and fair comparison.
>
> The setting of the variant effect prediction of eQTL tasks also differs significantly. In LRB, the task is to predict whether a SNP is a positive QTL, while DNALongBench aims to predict positive eQTL-gene pairs. Additionally, LRB combines data across all tissues, whereas DNALongBench separates datasets by tissue to better capture biologically meaningful tissue-specific regulatory interactions. The GTEx paper [1] demonstrated that eQTLs exhibit consistent tissue specificity (Figure 3a), supporting our approach of treating eQTL from different tissues as distinct datasets.
>
> In conclusion, while we acknowledge that some overlap exists with previous benchmarks, we believe **DNALongBench offers a fundamentally more comprehensive and scalable framework for evaluating long-range genomic tasks**. By including larger datasets, longer input sequences, and task types that directly address biologically more meaningful gene regulatory dynamics, DNALongBench fills critical gaps left by BEND and LRB and would be very useful for the field of genomic LM development. We hope this clarifies our contributions and demonstrates the unique value of DNALongBench. We remain open to any additional questions or feedback and greatly appreciate the reviewer's insights, which have been invaluable in strengthening this work.
>
> **Table R1. Number of tracks used by DNALongBench and LRB**
>
> |           | DNALongBench (Human) | DNALongBench (Mouse) | LRB |
> |-----------|:--------------------:|:--------------------:|:---:|
> |    CAGE   |          683         |          357         |  50 |
> |  CHIP-seq |         3991         |         1058         |  0  |
> | DNase-seq |          674         |          101         |  0  |
> |  ATAC-seq |           0          |          127         |  0  |
>
> [1] GTEx Consortium (2017). Genetic effects on gene expression across human tissues. Nature, 550(7675), 204-213

---

> > ### Author Response · Authors · 2024-11-28
> > **Response with additional contact map dataset and the corresponding results**
> >
> > We sincerely appreciate the reviewer for pointing out their further concerns. We are pleased to update the final evaluation results for the additional dataset curated for contact map prediction. The results are shown in the following table. Akita outperforms other methods across all four cell types, consistent with our observations in the five datasets initially used in the Akita paper. **In our initial experiments, we tried using the HUVEC Hi-C data. However, we found this dataset has very low coverage (around 10%) compared to other cell lines, leading to lower performance on the test dataset. We then chose another cell line, HepG2, and used another high-coverage Hi-C data for HeLa cells. We believe these newly curated datasets will be quite useful for others to thoroughly evaluate genomic models’ performance in predicting cell type-specific contact maps.**  We will also generate more datasets upon request to further expand the size of benchmarks. **We have correspondingly updated the text of the manuscript and tables to reflect this update at line 817-822, Table A10 and A11**.
> >
> > |                      | HAP1   | Hela  | HepG2 | K562  | Average |
> > |----------------------|--------|-------|-------|-------|---------|
> > | Expert model (Akita) | **0.196**  | **0.223** | **0.198** | **0.175** | **0.198**   |
> > | CNN                  | 0.018  | 0.025 | 0.021 | 0.003 | 0.017   |
> > | HyenaDNA             | -0.062 | 0.103 | 0.094 | 0.065 | 0.049   |
> > | Caduceus-Ph          | 0.063  | 0.168 | 0.178 | 0.003 | 0.103   |
> > | Caduceus-PS          | 0.063  | 0.170 | 0.178 | 0.051 | 0.115   |

---

### Meta-Review · Area_Chair_ashk · 2024-12-19

**Metareview:**

This is a well-executed DNA benchmark paper. The reviewers were pretty well-aligned on their score in the weak reject to accept range. Compared to previous work like BEND - published at last year's ICLR - this paper adds five long range benchmarks.

This work definitely deserved publication. But given its incremental nature, arguably a specialized (bioinformatics) venue is a more obvious choice than ICLR.

**Additional Comments On Reviewer Discussion:**

None.

---

### Decision · Program_Chairs · 2025-01-22

Reject